# Maintenance of homeostatic plasticity at the *Drosophila* neuromuscular synapse requires continuous IP$_3$-directed signaling

Thomas D James[1,2], Danielle J Zwiefelhofer[1], C Andrew Frank[1,3]*

[1]Department of Anatomy and Cell Biology, University of Iowa Carver College of Medicine, Iowa City, United States; [2]Interdisciplinary Graduate Program in Neuroscience, University of Iowa, Iowa City, United States; [3]Interdisciplinary Programs in Neuroscience, Genetics and Molecular Medicine, University of Iowa, Iowa City, United States

**Abstract** Synapses and circuits rely on neuroplasticity to adjust output and meet physiological needs. Forms of homeostatic synaptic plasticity impart stability at synapses by countering destabilizing perturbations. The *Drosophila melanogaster* larval neuromuscular junction (NMJ) is a model synapse with robust expression of homeostatic plasticity. At the NMJ, a homeostatic system detects impaired postsynaptic sensitivity to neurotransmitter and activates a retrograde signal that restores synaptic function by adjusting neurotransmitter release. This process has been separated into temporally distinct phases, induction and maintenance. One prevailing hypothesis is that a shared mechanism governs both phases. Here, we show the two phases are separable. Combining genetics, pharmacology, and electrophysiology, we find that a signaling system consisting of PLCβ, inositol triphosphate (IP$_3$), IP$_3$ receptors, and Ryanodine receptors is required only for the maintenance of homeostatic plasticity. We also find that the NMJ is capable of inducing homeostatic signaling even when its sustained maintenance process is absent.
**Editorial note:** This article has been through an editorial process in which the authors decide how to respond to the issues raised during peer review. The Reviewing Editor's assessment is that all the issues have been addressed (see decision letter).
DOI: https://doi.org/10.7554/eLife.39643.001

*For correspondence:
andy-frank@uiowa.edu

**Competing interests:** The authors declare that no competing interests exist.

## Introduction

Synaptic plasticity is a fundamental property of neurons that underlies the activities of neuronal circuits and behaviors. Neurons have a remarkable capacity to adjust outputs in response to external cues. Depending upon context, those adjustments can be stabilizing or destabilizing to overall function. Hebbian forms of neuroplasticity are generally thought to promote destabilizing changes. A great deal is known about the molecular mechanisms underlying Hebbian paradigms of synaptic plasticity like Long-Term Potentiation (LTP, e.g. *Lisman et al., 2012*) – and how Hebbian plasticity might underlie long-lasting processes like memory formation and consolidation (*Andersen et al., 2017*; *Poo et al., 2016*). Less is understood about homeostatic forms of neuroplasticity, which work to stabilize synapse function and keep activity levels within an acceptable physiological range (*Davis, 2006*; *Davis and Müller, 2015*; *Delvendahl and Müller, 2019*; *Pozo and Goda, 2010*; *Turrigiano, 2017*). For homeostatic plasticity, it is generally thought that coordinated actions of neurons and their targets work to maintain a set point functional parameter.

Well-studied examples of homeostatic synaptic plasticity (HSP) include synaptic scaling (*O'Brien et al., 1998*; *Turrigiano et al., 1998*; *Turrigiano and Nelson, 2004*), and the maintenance of evoked excitation at neuromuscular junctions (NMJs) (*Cull-Candy et al., 1980*; *Davis and Müller,*

*2015*; *Frank, 2014a*; *Petersen et al., 1997*). For both, the time course of implementation has been of longstanding interest. Synaptic scaling was initially shown to be a slow, chronically executed process (*O'Brien et al., 1998*; *Turrigiano et al., 1998*), but it is also possible for faster scaling mechanisms to be mobilized if multiple synaptic sites are concurrently inhibited (*Sutton et al., 2006*). For the NMJ, homeostatic signaling is triggered by short-term challenges to synapse function (*Frank et al., 2006*; *Wang et al., 2016b*), but it is also maintained for extended developmental time in the face of chronic challenges (*Cull-Candy et al., 1980*; *Davis et al., 1998*; *DiAntonio et al., 1999*; *Petersen et al., 1997*; *Plomp et al., 1992*).

The *Drosophila melanogaster* NMJ is an ideal model synapse for studying the basic question of how synapses work to counter destabilizing perturbations (*Frank, 2014a*). At this NMJ, reduced sensitivity to single vesicles of glutamate initiates a retrograde, muscle-to-nerve signaling cascade that induces increased neurotransmitter vesicle release, or quantal content (QC). As a result, the NMJ maintains a normal postsynaptic response level (*Frank et al., 2006*; *Petersen et al., 1997*). Mechanistically, this increase in QC depends upon the successful execution of discrete presynaptic events, such as increases in neuronal $Ca^{2+}$ influx and an increase in the size of the readily releasable pool (RRP) of synaptic vesicles (*Frank et al., 2006*; *Müller and Davis, 2012*; *Müller et al., 2012*). The field has termed this compensatory signaling process as presynaptic homeostatic potentiation (PHP) (*Delvendahl and Müller, 2019*). Two factors that govern the expression of PHP are the nature of the NMJ synaptic challenge and the amount of time elapsed after presentation of the challenge. Acute pharmacological inhibition of postsynaptic glutamate receptors initiates a rapid induction of PHP that restores synaptic output in minutes (*Frank et al., 2006*). By contrast, genetic lesions and other long-term reductions of NMJ sensitivity to neurotransmitter induce PHP in a way that is sustained throughout life (*Brusich et al., 2015*; *Davis et al., 1998*; *DiAntonio et al., 1999*; *Paradis et al., 2001*; *Petersen et al., 1997*).

We previously identified the *Plc21C* gene as a factor needed for PHP (*Brusich et al., 2015*). *Plc21C* encodes a *Drosophila* Phospholipase Cβ (PLCβ) homolog known to be neuronally expressed (*Shortridge et al., 1991*) – but recent ribosomal profiling data also indicates possible muscle expression of *Plc21C* (*Chen and Dickman, 2017*). In canonical signaling pathways, once PLCβ is activated by Gαq, it cleaves the membrane lipid phosphatidylinositol 4,5-bisphosphate ($PIP_2$) into diacylglycerol (DAG) and inositol triphosphate ($IP_3$). DAG can affect synaptic function by activating Protein Kinase C (PKC), while $IP_3$ binds its receptor ($IP_3$R) to trigger release of calcium from intracellular stores (*Kadamur and Ross, 2013*; *Philip et al., 2010*; *Tedford and Zamponi, 2006*). It is not understood which aspects of this signaling machinery are mobilized during PHP. Potential downstream consequences of PLCβ activity at the NMJ include phosphorylation of neuronal proteins, modulation of ion channel activity, and changes in localization of neurotransmission machinery (*Cremona and De Camilli, 2001*; *Goñi and Alonso, 1999*; *Huang et al., 2006*; *Peters et al., 2001*; *Rohrbough and Broadie, 2005*; *Wu et al., 2002*).

For this study, we scrutinized PLCβ-directed signaling further. We tested whether PLCβ-directed signaling was required solely for the maintenance of PHP or if it could also be required for induction. In addition to PLCβ, we identified the $IP_3$ Receptor (*Drosophila* Itpr, herein $IP_3$R) and Ryanodine receptor (*Drosophila* RyR) as being part of the same signaling process. We found that neither PLCβ, nor $IP_3$R, nor RyR are required for the rapid induction of PHP. Additionally, we found that the rapid induction of PHP is still possible in synapses already sustaining PHP. Surprisingly, we found that NMJs are capable of rapidly inducing PHP – even when the sustained expression of PHP is already blocked by impairments in PLCβ, $IP_3$R, or RyR signaling. Taken together, our data show that the induction and maintenance of PHP are separable. Even though there is compelling evidence that parts of the induction and maintenance signaling mechanisms overlap (*Goel et al., 2017*), it is also true that acute PHP is possible in scenarios where long-term PHP is not.

## Results

### PLCβ loss uncouples the short-term induction of homeostatic plasticity from its long-term maintenance

Previously, we demonstrated that loss of function of *Plc21C*, a *Drosophila melanogaster* PLCβ gene, could dampen or eliminate the long-term maintenance of PHP (*Brusich et al., 2015*). We repeated

some of those experiments. We used a fruit fly line containing both neuron- and muscle-GAL4 drivers as well as a *UAS-GluRIII[RNAi]* transgenic construct to provide a chronic homeostatic challenge to reduce quantal size (*Brusich et al., 2015*). Pre-+Post Gal4>>UAS-GluRIII[RNAi] NMJs have decreased quantal size (mEPSP, *Figure 1A*) and an offsetting, homeostatic increase in quantal content (QC, *Figure 1C*). This increase in release keeps excitatory postsynaptic potentials (EPSPs) at

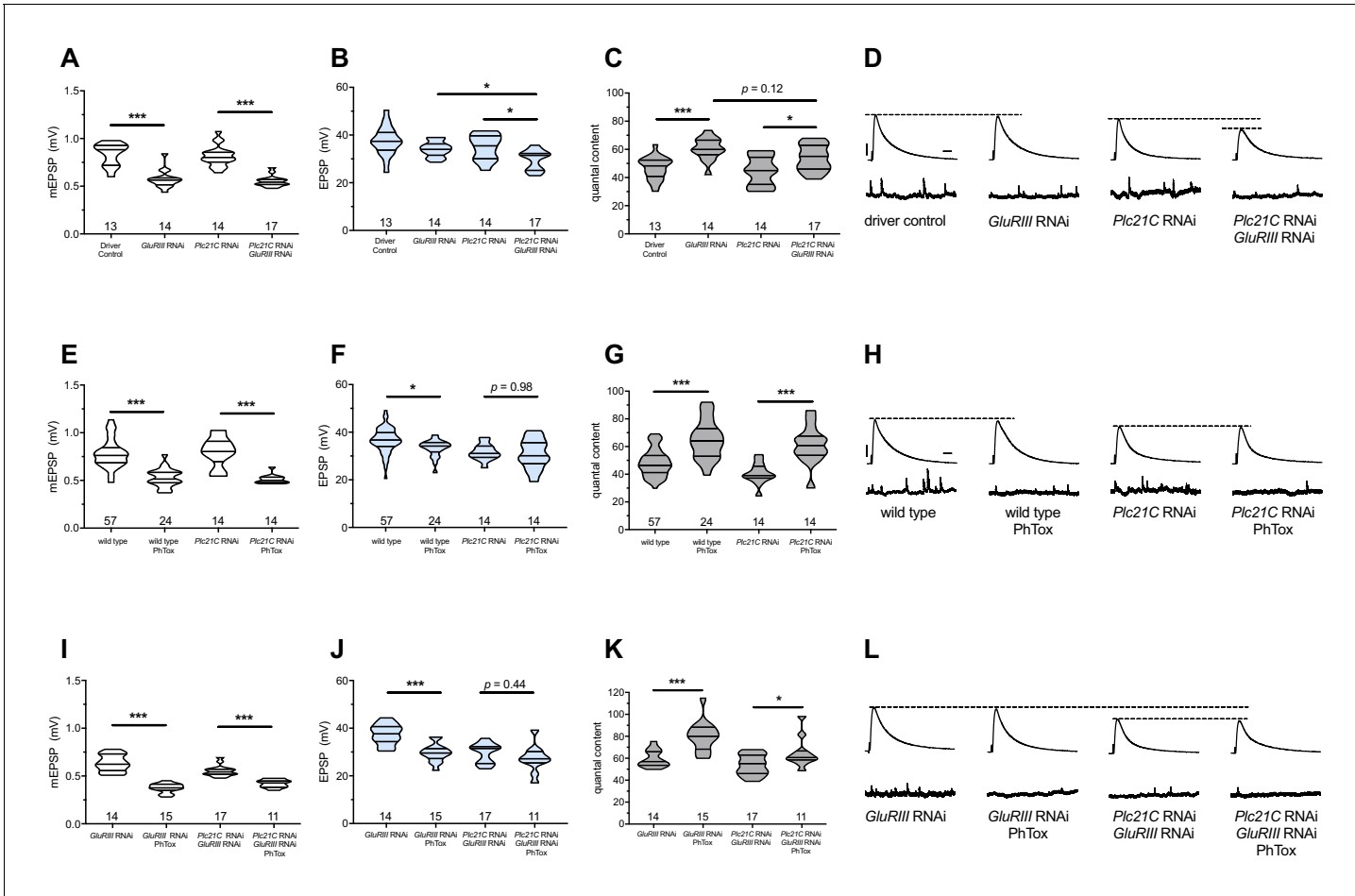

**Figure 1.** Maintenance of presynaptic homeostatic potentiation requires PLCβ, but induction does not. (A) *GluRIII* knockdown induces a significant decrease in quantal size for both driver control and *Plc21C* knockdown genetic backgrounds. (B) EPSP amplitudes are maintained with *GluRIII* knockdown alone but dampened with concurrent *GluRIII* and *Plc21C* knockdown. (C) There is significant PHP (Presynaptic Homeostatic Potentiation, measured as an increase in quantal content) in response to *GluRIII* knockdown. (D) Representative electrophysiological traces of EPSPs (above) and mEPSPs (below). The evoked events show full PHP in the *GluRIII* RNAi knock down background and partial PHP with concurrent *Plc21C* knock down. (E) 10-min incubation with 20 μM PhTox diminishes quantal size for all conditions. (F) EPSP amplitudes after acute PhTox incubation are maintained at or near normal levels. (G) The data in (F) are because PHP is rapidly induced in wild-type and *Plc21C* RNAi NMJs after PhTox incubation. (H) Representative electrophysiological traces show fully intact PHP induction. (I) With dual PHP maintenance (GluRIII knockdown) and induction (PhTox application), quantal size is further decreased. (J) EPSP amplitudes after dual maintenance and induction. (K) Quantal is content further increased in by PhTox treatment in *GluRIII* RNAi synapses compared to untreated synapses; this expression of PHP does not require full PLCβ function. (L) Representative electrophysiological traces illustrate that even though PHP maintenance is impaired with PLCβ knockdown, PHP induction is not impaired. Violin plots have horizontal lines signifying the 0th, 25th, 50th, 75th, and 100th percentiles of the data distribution; the distribution itself is delineated by the shapes of the plots. *p<0.05, **p<0.01, ***p<0.001 by Student's T-Test versus non-challenged genetic control or by one-way ANOVA with a Bonferroni post-hoc test in the case of comparing degree of compensation with *GluRIII* RNAi and *Plc21C* RNAi + *GluRIII* RNAi. Scale bars for all traces are y = 10 mV (1 mV), x = 20 ms (500 ms) for EPSPs (mEPSPs).

DOI: https://doi.org/10.7554/eLife.39643.002

The following source data is available for figure 1:

**Source data 1.** Raw electrophysiology data for *Figure 1*.
DOI: https://doi.org/10.7554/eLife.39643.003

control levels (*Figure 1B*). By contrast, concurrent knockdown of *GluRIII* and *Plc21C* gene functions by RNAi (*Pre-+Post* Gal4>>*UAS-GluRIII[RNAi]+Plc21C[RNAi]*) leaves this form of homeostatic potentiation only partly intact (*Figure 1A–C*). There is a small QC increase compared to baseline (*Figure 1C*), but this QC increase is blunted compared to the homeostatic challenge, resulting in evoked potentials that are smaller than controls (*Figure 1B and D*). These data are consistent with the prior results (*Brusich et al., 2015*) (*Supplementary file 1* for summary *Figure 1* data) (*Figure 1—source data 1* for raw data).

The *GluRIII* RNAi knock down manipulation in muscle is a days-long, chronic homeostatic challenge to the maintenance of NMJ function (*Brusich et al., 2015*). We tested if *Plc21C* gene knock down blocks or impairs the acute induction of PHP. For acute induction, we applied 20 µM of the glutamate receptor antagonist Philanthotoxin-433 (PhTox) to both wild-type and to *Pre-+Post* Gal4>>*Plc21C[RNAi]* knock down NMJs. PhTox application decreased quantal size for both conditions (*Figure 1E*) (*Frank et al., 2006*). For both conditions, evoked potentials remained largely steady compared to non-PhTox controls (*Figure 1F and H*) because there was a significant, compensatory increase in quantal content (*Figure 1G*). Thus, partial loss of *Plc21C* gene function is not a sufficient condition to block the rapid induction of PHP.

## The induction of PHP is possible, even when PHP maintenance is impaired

We used *Plc21C* loss and PhTox to test whether the capacity to maintain PHP for extended developmental time is required for rapid PHP induction. The most common modes of assessing PHP at the *Drosophila* NMJ are a lifelong, genetic *GluRIIA*$^{SP16}$ null mutation for PHP maintenance (*Petersen et al., 1997*) and acute PhTox application for PHP induction (*Frank et al., 2006*). For both cases, mEPSP amplitudes are decreased and QC is increased helping to maintain evoked potentials at (or nearly at) normal levels. PhTox targets the function of GluRIIA-containing receptors; thus, adding PhTox to a *GluRIIA*$^{SP16}$ null background does not further decrease quantal size (*Frank et al., 2006*). This fact presents a difficulty in using PhTox and *GluRIIA*$^{SP16}$ together to test whether additional PHP can be acutely induced in a chronic glutamate receptor loss genetic condition already sustaining PHP. We reasoned that by applying PhTox to *UAS-GluRIII[RNAi]* knock down synapses, we could circumvent this limitation. Partial loss of the essential subunit-encoding *GluRIII* gene leaves some GluRIIA-containing receptors intact (*Brusich et al., 2015*). In turn, those GluRIIA-containing receptors could be subject to the secondary PhTox challenge.

We applied PhTox to *UAS-GluRIII[RNAi]* synapses, and we observed a further decrease in quantal amplitude – significantly below mEPSP size recorded for *UAS-GluRIII[RNAi]* alone (*Figure 1I*). Evoked potentials were only slightly lower than the non-PhTox levels (*Figure 1J,L*) because there was a robust increase in QC (*Figure 1K*). This result indicated that a rapid induction of PHP was possible at a synapse already undergoing a sustained maintenance of PHP.

We next tested whether compromised ability to sustain PHP throughout life would also preclude acute induction of PHP. PhTox applied to NMJs simultaneously expressing both *UAS-GluRIII[RNAi]* and *Plc21C[RNAi]* constructs induced a significant decrease in mEPSP amplitude relative to non-PhTox-treated control synapses (*Figure 1I*). Yet we also observed a significant increase in QC (i.e. PHP induction) (*Figure 1K*), which kept evoked NMJ potentials similar to their non-PhTox levels (*Figure 1J,L*). Collectively, these data suggest that acute PHP induction does not require intact PHP maintenance and that PLCβ plays a maintenance role.

## IP$_3$ function is required for the maintenance of PHP but not its induction

We sought to identify potential PLCβ signaling effectors that could mediate the long-term maintenance of PHP. We screened targets by electrophysiology. Based on canonical signaling functions of PLCβ, we conducted a directed screen, targeting molecules such as PKC, CaMKII, Unc-13, related signaling molecules, as well several potential G-protein-coupled receptors (GPCRs). Additionally, we tested molecules implicated in intracellular calcium signaling, intracellular ion channel function, and synaptic ion channel function.

We used a screening paradigm designed to find factors needed for the maintenance of PHP: combining pre- and postsynaptic GAL4 expression with *UAS-GluRIII[RNAi]* with a tested genetic

manipulation (*Brusich et al., 2015*). We targeted factors for this screen using either *UAS-gene* mis-expression or *UAS-gene[RNAi]* constructs. We also used loss-of-function mutations. For an additional screening condition for some mutations, we constructed double mutant lines with a *GluRIIA^{SP16}* null deletion allele (*Petersen et al., 1997*). For any test, we analyzed two conditions: a baseline neurotransmission condition (e.g. GAL4 +genetic manipulation alone) and a homeostatically challenged condition (e.g. GAL4 +genetic manipulation+*UAS-GluRIII[RNAi]*). Any homeostatically challenged condition that failed to increase QC over its own baseline condition was designated as a potential positive.

We examined 28 distinct genetic manipulations (comprising 23 distinct genes), including controls (*Figure 2A*) (*Supplementary file 2* for summary *Figure 2* data) (*Figure 2—source data 1* for raw data). We plotted the relative QC values for the screen as '% baseline' (*Figure 2A*), indicating how much of a QC change the *UAS-GluRIII[RNAi]* challenge yielded. We set a cutoff for a 'screen positive' as a QC smaller than one standard deviation below the expected QC given the *UAS-GluRIII [RNAi]* homeostatic challenge (*Figure 2A*, red dashed line).

Two genetic manipulations showed no statistically significant QC increase upon homeostatic challenge (*Figure 2A*, red). For both manipulations, the screened target molecule was inositol 1,4,5-triphosphate ($IP_3$). $IP_3$ is a second-messenger signaling molecule. We examined it because PLCβ cleaves the phospholipid $PIP_2$ into soluble $IP_3$ and membrane-bound diacylglycerol (DAG) during canonical signaling. We targeted cellular $IP_3$ by expressing *UAS-IP$_3$-sponge*, a transgene that expresses a peptide that binds and sequesters $IP_3$ (*Usui-Aoki et al., 2005*). Concomitant pre-and postsynaptic expression of *UAS-IP$_3$-sponge* transgenes completely blocked the long-term expression of PHP (*Figure 2B–E*). This resulted in significantly smaller EPSP amplitudes in the *UAS-GluRIII[RNAi]* PHP-challenge genetic background. (*Figure 2B,D*).

We used the full block of sustained PHP by *UAS-IP$_3$-sponge* expression to re-test the relationship between the rapid induction of PHP and its long-term maintenance. First, we tested if *UAS-IP$_3$-sponge* expression alone could block the rapid induction of PHP. Following PhTox treatment, synapses expressing *UAS-IP$_3$-sponge* in pre- and postsynaptic tissues showed a significant decrease in mEPSP amplitude compared to non-PhTox controls (*Figure 2F*), and yet they had steady EPSP amplitudes (*Figure 2G*) because of a robust increase in QC (*Figure 2H*). This result indicated that *UAS-IP$_3$-sponge* expression left the rapid induction mechanisms of PHP intact.

We next tested if the rapid induction of PHP was possible for third instar larval NMJs that had blocked PHP maintenance throughout life. To do this, we applied PhTox to NMJs expressing *UAS-IP$_3$-sponge* in a *UAS-GluRIII[RNAi]* background. Even in this genetic background, we found that the rapid induction of PHP remained intact following PhTox treatment (*Figure 2F–H*). PhTox treatment resulted in a further decrease in mEPSP amplitude compared to non-PhTox-treated synapses expressing both *UAS-GluRIII[RNAi]* and *UAS-IP$_3$-sponge* (*Figure 2F*), but there was an offsetting increase in QC (*Figure 2H*). The evoked event amplitudes were at the level of genetically identical, non-PhTox-treated synapses (*Figure 2G*), because of successful induction of PHP in a genetic background that was unable to sustain PHP throughout development.

## IP$_3$ sequestration does not impair synapse growth

Chronic expression of the *UAS-IP$_3$-sponge* transgene blocked the long-term expression of PHP. In principle, this result could be a secondary consequence of aberrant NMJ development. To check this possibility, we co-immunostained third instar larval *Drosophila* NMJs with anti-Synapsin (Syn, presynaptic vesicles), anti-Discs Large (Dlg, postsynaptic density), and anti-Horseradish Peroxidase (HRP, presynaptic membrane) antibodies. This allowed us to examine synaptic growth by counting NMJ boutons. We quantified bouton growth for synapse 6/7, muscle segments A2 and A3. We examined control conditions and conditions with blocked PHP maintenance due to *UAS-IP3-sponge* expression (*Figure 3A–E*). We observed no significant differences versus control in bouton number for any condition, for either segment A2 or A3 – including the genetic background where we co-expressed *UAS-IP3-sponge* and *UAS-GluRIII[RNAi]* (*Figure 3F*). There were also no significant differences versus control in bouton number normalized per unit muscle area (*Figure 3G*) (*Figure 3—source data 1* for raw bouton count and muscle size data). These data indicate that when $IP_3$ is sequestered, synapse undergrowth is not causal for a PHP block.

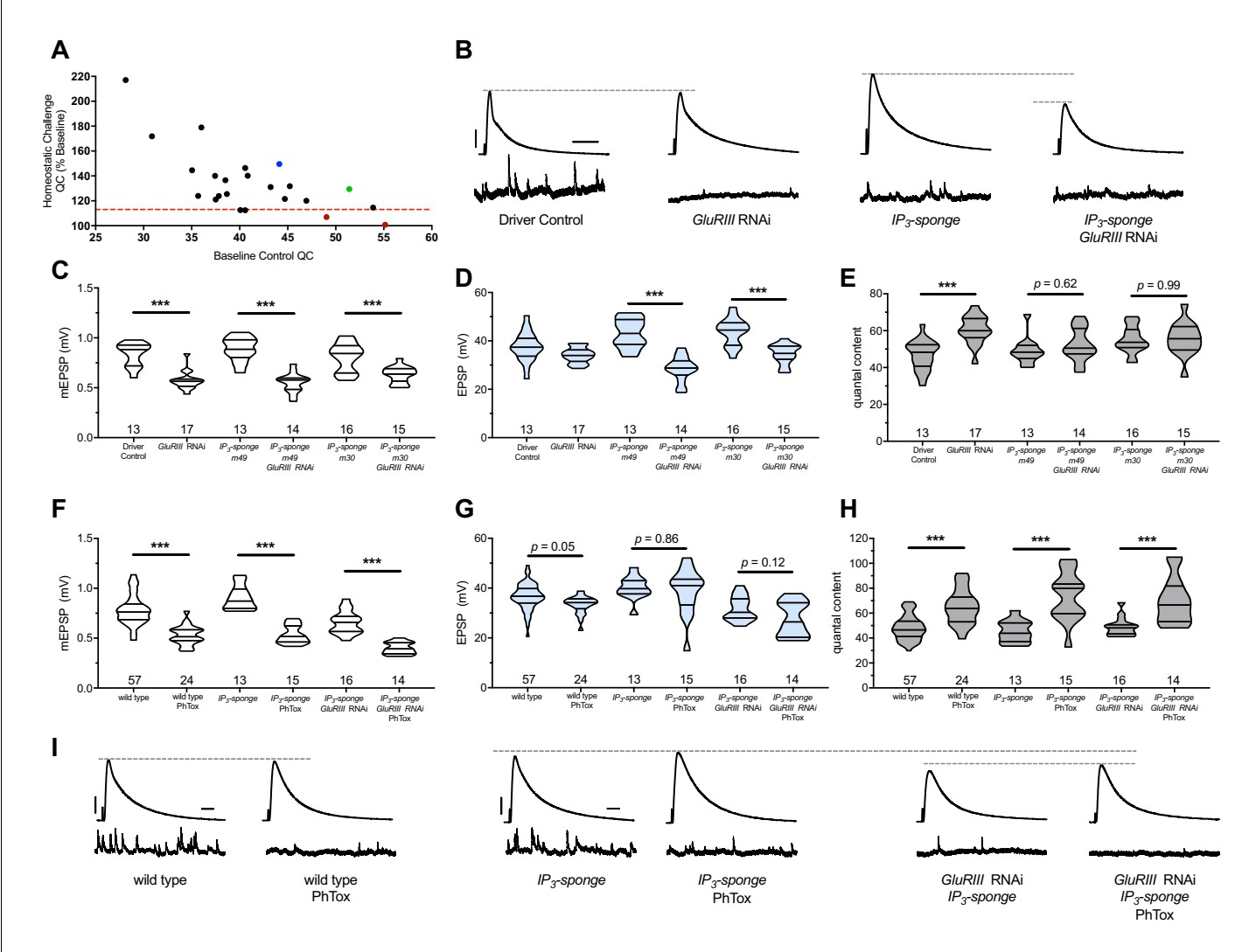

**Figure 2.** IP$_3$ sequestration blocks PHP maintenance but not PHP induction. (A) Screen data, plotting baseline quantal content (QC, *x*-axis, genetic manipulation alone) versus QC of the homeostatically challenged condition (*y*-axis, *GluRIII RNAi* or *GluRIIA* mutant). Blue = *GluRIIA* alone. Green = *GluRIII RNAi* alone. Red = *GluRIII RNAi+UAS-IP$_3$-sponge*. Dotted line = one standard deviation below the mean QC of *GluRIII* RNAi. (B) Representative electrophysiological traces (EPSPs above; mEPSPs below), demonstrating diminished evoked potentials in the *GluRIII RNAi +UAS-IP$_3$-sponge* (presynaptic +postsynaptic expression) condition. (C) *GluRIII* knockdown induces a significant decrease in quantal size for all genetic backgrounds. (D) EPSP amplitudes are maintained with *GluRIII* knockdown alone but significantly diminished with concurrent *GluRIII* knockdown and expression of either *UAS-IP$_3$-sponge* line. (E) By quantal content, sustained PHP expression is abolished when *UAS-IP$_3$-sponge* is expressed using concurrent pre- and postsynaptic GAL4 drivers. (F) 10-min incubation with 20 µM PhTox diminishes quantal size for all conditions. (G) EPSP amplitudes after acute PhTox incubation are maintained at or near normal levels for all conditions. (H) Failure to sustain PHP following IP$_3$ sequestration does not preclude its rapid induction. The data in (G) are because PhTox treatment results in a compensatory increase in QC for conditions shown. (I) Representative electrophysiological traces showing full homeostatic compensation with PhTox application (as in *Frank et al., 2006*). (J) Representative traces of the experimental data in (F–H). Violin plots used as in *Figure 1*. *p<0.05, **p<0.01, ***p<0.001 by Student's T-Test versus non-challenged genetic control. Scale bars for all traces are *y* = 10 mV (1 mV), *x* = 20 ms (500 ms) for EPSPs (mEPSPs).
DOI: https://doi.org/10.7554/eLife.39643.004

The following source data is available for figure 2:

**Source data 1.** Raw electrophysiology data for *UAS-IP$_3$-sponge* experiements in *Figure 2*.
DOI: https://doi.org/10.7554/eLife.39643.005

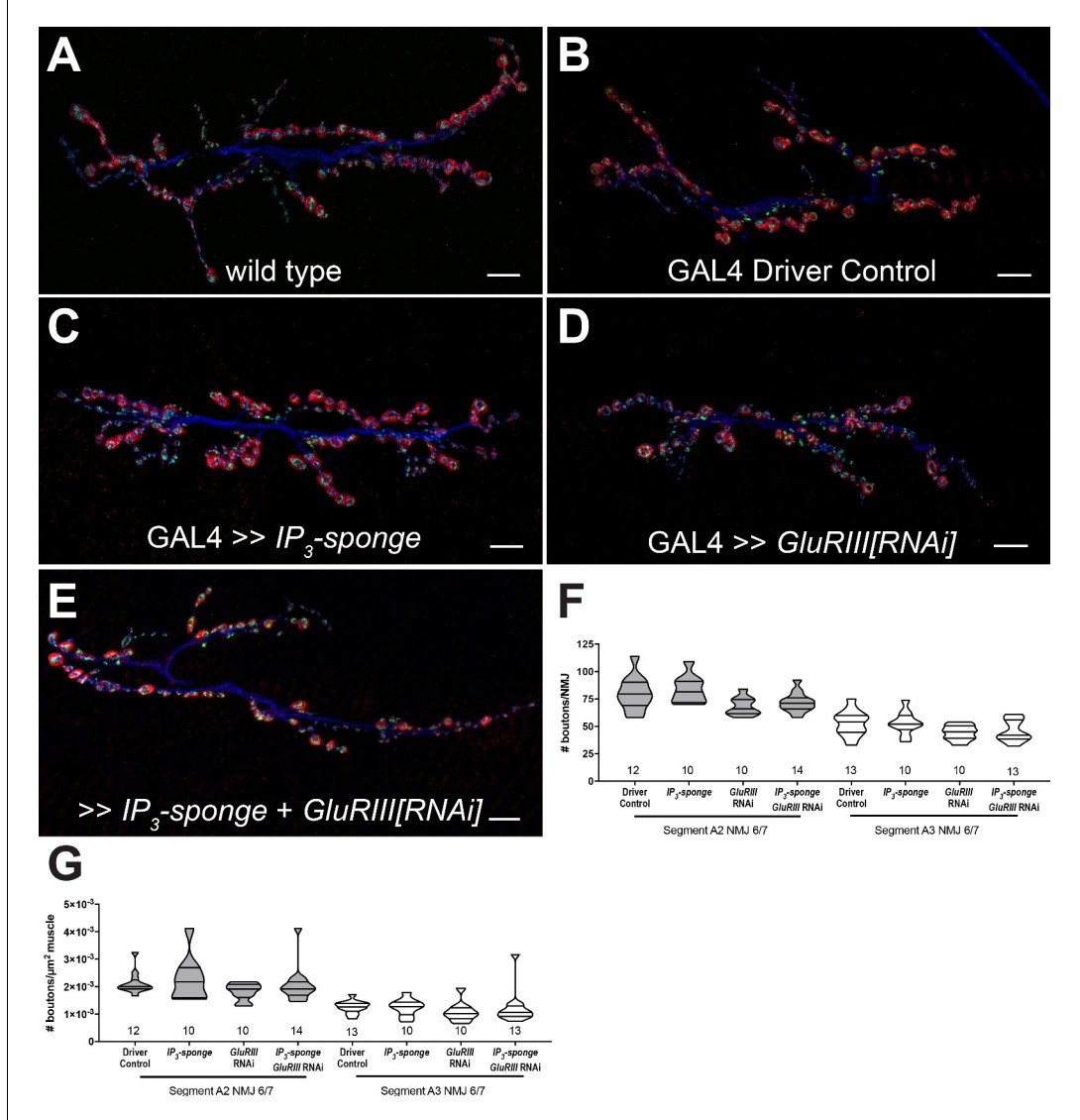

**Figure 3.** IP$_3$ sequestration does not impair NMJ growth. (A–E) NMJs were co-stained with anti-DLG (red) and anti-Synapsin antibodies (green) to visualize synaptic boutons, with anti-HRP (blue) to visualize presynaptic membranes. Genotypes or conditions as indicated. All scale bars, 10 µm. (F) NMJ growth was assessed by bouton counting at abdominal segments A2 and A3, muscle 6/7, based on postsynaptic DLG staining and double checked for presynaptic Synapsin. No statistically significant differences in NMJ growth versus driver control were observed for any of the experimental conditions (p>0.1 vs. control, regardless of segment). (G) Bouton counts were normalized per unit of muscle 6/7 area. No statistically significant differences versus control were observed (p>0.2 vs. control, regardless of segment). Violin plots used as in *Figure 1*. For both F and G, data were compared for each segment individually using the Kruskal-Wallis ANOVA test followed by Dunn's multiple comparisons test.
DOI: https://doi.org/10.7554/eLife.39643.006
The following source data is available for figure 3:

**Source data 1.** Raw synapse growth data for *Figure 3*.
DOI: https://doi.org/10.7554/eLife.39643.007

## Pharmacology targeting IP$_3$ receptors uncouples the induction and maintenance of PHP

We tested if the temporal requirements of PHP could be uncoupled by pharmacological disruption of *Drosophila* IP$_3$ receptor function (Itpr in *Drosophila*). IP$_3$Rs are localized to the endoplasmic reticulum (ER) and function to mediate calcium efflux from internal stores (*Berridge, 1984*; *Berridge, 1987*). ER is known to localize throughout neurons in *Drosophila*, including synaptic terminals

(*Summerville et al., 2016*). Recent studies have implicated ER resident proteins in the execution of PHP (*Genç et al., 2017*) or in baseline neurotransmission and synapse growth (*Kikuma et al., 2017*) at the *Drosophila* NMJ. To target IP$_3$Rs, we turned to two reagents known to impair function: Xestospongin C and 2-APB (2-Aminoethoxydiphenyl Borate) and applied those drugs to *GluRIIA* loss-of-function mutants.

Xestospongin C is a membrane-permeable drug that disrupts intracellular calcium release directly via non-competitive inhibition of IP$_3$Rs (*Gafni et al., 1997*; *Wilcox et al., 1998*). Xestospongin C has been previously shown to inhibit *Drosophila* IP$_3$Rs (*Vázquez-Martínez et al., 2003*). There are caveats to its use; Xestospongin C may act indirectly by inhibiting SERCA, which could lead to depletion of intracellular calcium stores (*Castonguay and Robitaille, 2002*). Moreover, Xestospongin C has been demonstrated to impair voltage-gated Ca$^{2+}$ and K$^+$ currents in guinea pig smooth muscle (*Ozaki et al., 2002*). In principle, these latter activities on intact fly NMJ tissue could impact baseline neurotransmission parameters (*Bergquist et al., 2010*; *Brusich et al., 2015*; *Jan et al., 1977*; *Peng and Wu, 2007*).

*GluRIIA$^{SP16}$* null mutant NMJs have a marked decrease in quantal size (*Figure 4A*) (*Petersen et al., 1997*). This homozygous null condition does not perfectly maintain control EPSP amplitudes (*Figure 4B*) (*Brusich et al., 2015*; *Frank et al., 2006*; *Frank et al., 2009*; *Spring et al., 2016*; *Yeates et al., 2017*). However, the null does induce a robust increase in QC, which signifies a long-term implementation of PHP (*Figure 4C*). *GluRIIA$^{SP16}$* NMJ preparations acutely treated with 20 µM Xestospongin C (10 min) displayed an expected decrease in mEPSP amplitude compared to non-*GluRIIA$^{SP16}$* controls (*Figure 4A*). However, these drug-treated NMJs failed to show an increase in QC (*Figure 4C*). This resulted in markedly decreased evoked amplitudes (*Figure 4B,G*). Thus, not only did 20 µM Xestospongin C induce a block of PHP maintenance, but it was capable of extinguishing this long-term maintenance process on a timescale of minutes.

Importantly, 20 µM Xestospongin C did not impair baseline neurotransmission in a wild-type background (*Figure 4B,G*); this suggests that 20 µM Xestospongin C does not impair the function of other important voltage-gated channels at the *Drosophila* NMJ. Finally, in the *GluRIIA* null background, neither a lower dose (5 µM Xestospongin C) nor vehicle application alone inhibited the expression of PHP (*Figure 4A–C*) (*Supplementary file 3* for summary *Figure 4* data) (*Figure 4—source data 1* for raw data).

Next, we tested if acute application of 20 µM Xestospongin C could block the rapid induction of PHP. We applied 20 µM Xestospongin C to wild-type NMJs concurrently with 20 µM PhTox. Quantal size was markedly diminished compared to the non-PhTox control (*Figure 4D*; vehicle control dataset same as in 4A-C). Yet evoked amplitudes remained near control levels (*Figure 4E,G*) because the rapid induction of PHP was intact (*Figure 4F*).

We used 2-APB as a second reagent to target IP$_3$Rs. 2-APB is a membrane-permeable drug that has variable effects. It is known to impair IP$_3$R (*Diver et al., 2001*; *Maruyama et al., 1997*). There are also reports that 2-APB can impair targets such as Transient Receptor Potential (TRP) channels (*Bootman et al., 2002*; *Xu et al., 2005*). If continuous IP$_3$R function were required for the maintenance PHP at NMJs, we reasoned that acute application of 2-APB (as with Xestospongin C) should also extinguish this form of neuroplasticity.

We applied both 1 µM and 10 µM 2-APB to *GluRIIA$^{SP16}$* null NMJs. Both drug concentrations resulted in a failure to increase QC compared to drug-treated wild-type controls; this resulted in small evoked events for the drug-treated *GluRIIA$^{SP16}$* NMJs because PHP maintenance was blocked (*Figure 5A–C,G*) (*Supplementary file 4* for summary *Figure 5* data) (*Figure 5—source data 1* for raw data). Next, we tested if 2-APB blocks the PhTox-induced rapid induction of PHP. We applied 1 µM 2-APB concurrently with 20 µM PhTox. Evoked potentials remained near the level of 2-APB-treated NMJs without PhTox (*Figure 5E,H*) because 1 µM 2-APB left the rapid induction of PHP intact (*Figure 5D–F*).

We note that 2-APB potentiated baseline neurotransmission, seemingly in a dose-dependent way (*Figure 5B,G*). This potentiation likely means that 2-APB had off-target effects at the NMJ in addition to IP$_3$Rs. We considered that 2-APB could exert effects though TRP channels, like Drosophila Inactive (Iav). Iav plays a role in NMJ neurotransmission and controls Ca$^{2+}$ levels in motor neurons (*Wong et al., 2014*). However, knock down of *iav* gene function by RNAi did not impair PHP in our screen (*Supplementary file 2*), and the effects of 2-APB on baseline neurotransmission appear to be the opposite of those reported for strong *iav* loss of function (*Wong et al., 2014*). Taken together,

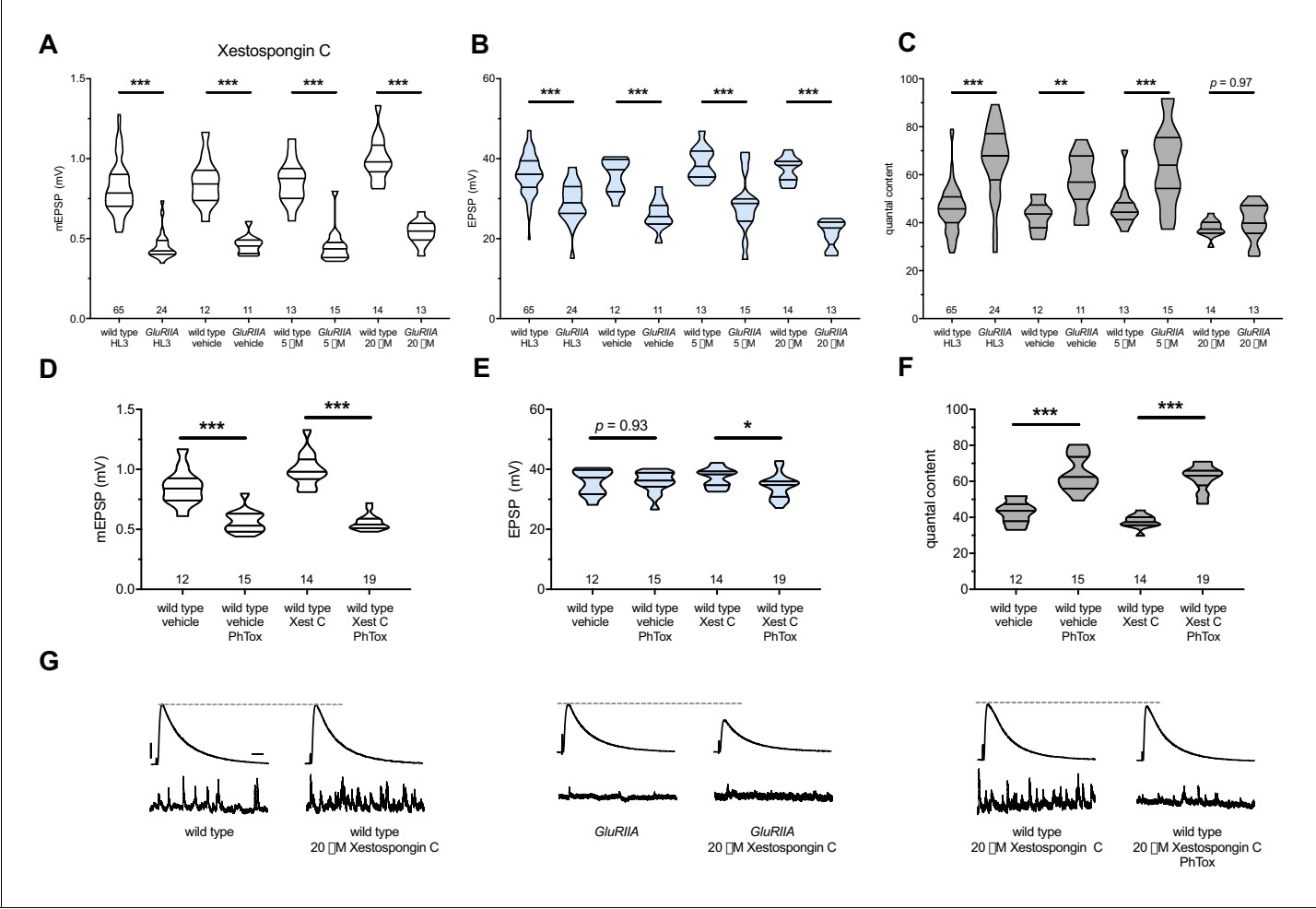

**Figure 4.** Xestospongin C blocks PHP maintenance but not PHP induction. Xestospongin C acutely applied to NMJs to impair IP$_3$R function. (A) The *GluRIIA$^{SP16}$* deletion mutation diminishes quantal size for all experimental conditions. (B) EPSP amplitudes are somewhat impaired versus non-*GluRIIA* control in all cases but most severely impaired when *GluRIIA* deletion is combined with 20 μM Xestospongin C incubation. (C) By quantal content, sustained PHP expression is abolished with acute NMJ exposure to 20 μM Xestospongin C. (D) Acute incubation 20 μM PhTox diminishes quantal size for all conditions (PhTox +DMSO vehicle or PhTox +20 μM Xestospongin C). (E) EPSP amplitudes are normal or near normal for all conditions. (F) Rapid PHP induction by PhTox not blocked by 20 μM Xestospongin C. (G) Representative electrophysiological traces (EPSPs above; mEPSPs below) for 20 μM Xestospongin C (or control) conditions in (A–F). Violin plots used as in *Figure 1*. *p<0.05, **p<0.01, ***p<0.001 by Student's T-Test versus non-challenged genetic control. Scale bars for all traces are *y* = 10 mV (1 mV), *x* = 20 ms (500 ms) for EPSPs (mEPSPs).
DOI: https://doi.org/10.7554/eLife.39643.008

The following source data is available for figure 4:

**Source data 1.** Raw electrophysiology data for *Figure 4*.
DOI: https://doi.org/10.7554/eLife.39643.009

our pharmacological data show that with acute drug application of either Xestospongin C or 2-APB, it is acutely possible to erase a lifelong, *GluRIIA$^{SP16}$*-induced long-term expression of PHP. Since this erasure is accomplished with known inhibitors of IP$_3$R, our data are consistent with the hypothesis that the maintenance of PHP requires continuous IP$_3$R function.

## Pharmacology targeting ryanodine receptors uncouples the induction and maintenance of PHP

Ryanodine receptors (RyRs) also mediate release of calcium from ER stores (*Berridge, 1998*; *Simkus and Stricker, 2002*). RyRs are localized to the ER in excitable tissues like neurons and muscle (*Santulli and Marks, 2015*; *Santulli et al., 2017*). Therefore, we tested whether RyRs are also

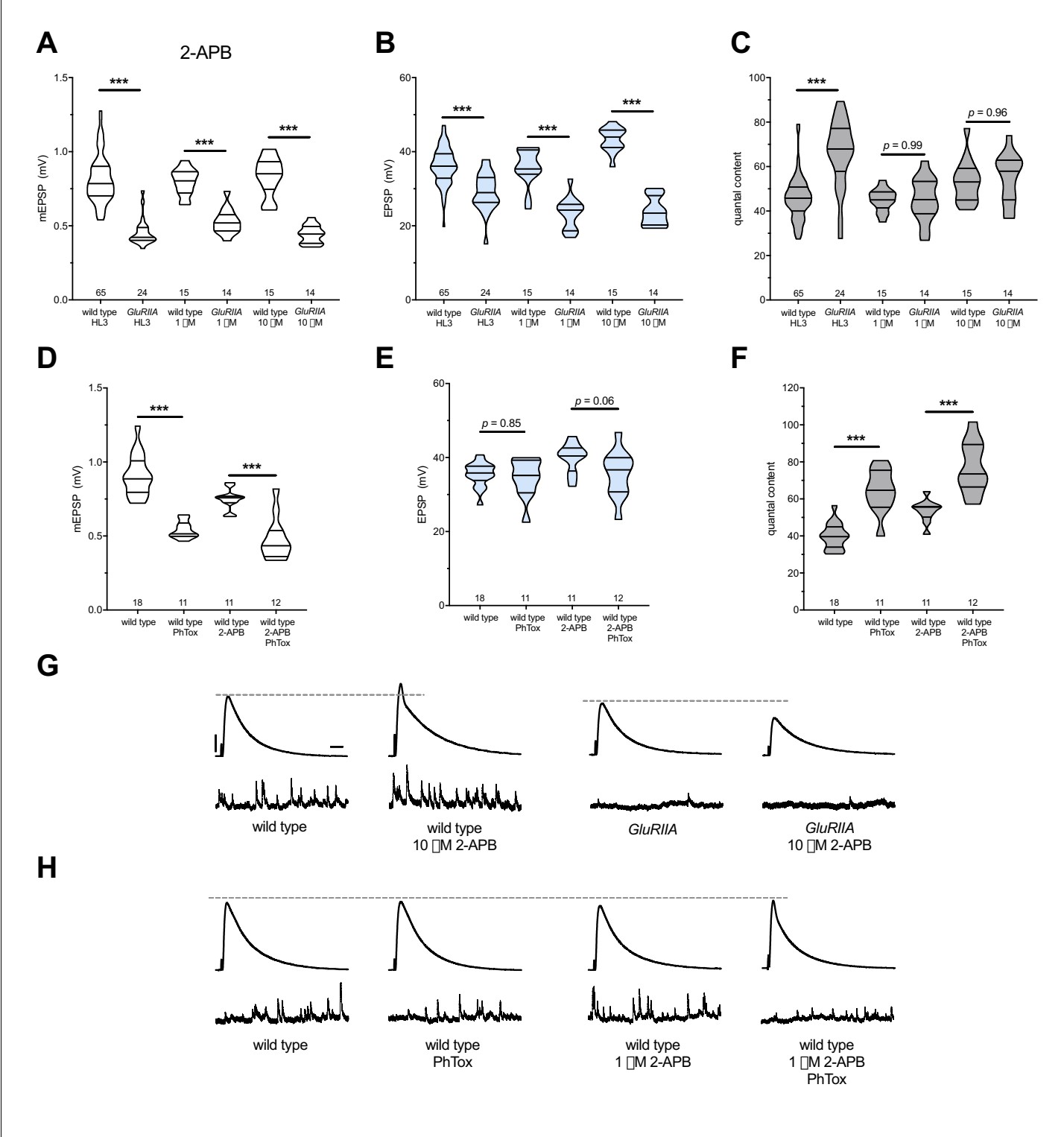

**Figure 5.** 2-APB blocks PHP maintenance but not PHP induction. 2-APB acutely applied to NMJs to impair IP$_3$R function. Wild-type and *GluRIIA* control data sets are replotted from *Figure 4* for visual comparison. (**A**) The *GluRIIA$^{SP16}$* deletion mutation diminishes quantal size for all experimental conditions. (**B**) EPSP amplitudes are somewhat impaired versus non-*GluRIIA* control in all cases but severely impaired when *GluRIIA* deletion is combined with either 1 µM or 10 µM 2-APB incubation. (**C**) By quantal content, sustained PHP expression is abolished with acute NMJ exposure to 1 µM or 10 µM 2-APB. (**D**) Acute incubation 20 µM PhTox diminishes quantal size for all conditions (PhTox +DMSO vehicle or PhTox +1 µM 2-APB). (**E**) EPSP amplitudes are normal or near normal for all conditions. (**F**) Rapid PHP induction by PhTox not blocked by 1 µM 2-APB. (**G**) Representative

*Figure 5 continued*

electrophysiological traces (EPSPs above; mEPSPs below) for 10 µM 2-APB (or control) conditions in (**A-C**). (**H**) Representative electrophysiological traces for 1 µM 2-APB (or control) conditions in (**D-F**). Violin plots used as in *Figure 1*. \*p<0.05, \*\*p<0.01, \*\*\*p<0.001 by Student's T-Test versus non-challenged genetic control. Scale bars in (**G**) also apply to (**H**) and are $y$ = 10 mV (1 mV), $x$ = 20 ms (500 ms) for EPSPs (mEPSPs).

DOI: https://doi.org/10.7554/eLife.39643.010

The following source data is available for figure 5:

**Source data 1.** Raw electrophysiology data for *Figure 5*.

DOI: https://doi.org/10.7554/eLife.39643.011

required for the maintenance of PHP at the NMJ. We repeated the same types of experiments executed with the IP$_3$R pharmacological blockade – this time targeting *Drosophila* RyRs by utilizing Ryanodine (*Murmu et al., 2010*) and Dantrolene (*Vázquez-Martínez et al., 2003*; *Zhao et al., 2001*) at concentrations previously reported to block RyRs. We acquired similar results as with IP$_3$R blockade. Acute application of either 100 µM Ryanodine or 10 µM Dantrolene to *GluRIIA* null preparations resulted in failure of PHP maintenance. QC did not increase for the homeostatically challenged condition (*GluRIIA* +drug) versus the unchallenged condition (wild-type +drug) (*Figure 6A–C*) (*Supplementary file 5* for summary *Figure 6* data) (*Figure 6—source data 1* for raw data).

Next, we tested whether RyR inhibition could block the rapid induction of PHP. As was the case with IP$_3$R inhibition, acute application of 10 µM Dantrolene to PhTox-treated NMJs did not disrupt the short-term induction of PHP (*Figure 6D–F*). We extended our analysis by re-examining the condition where we combined a long-term homeostatic challenge (*GluRIII[RNAi]*) with a short-term challenge (PhTox) to further decrease quantal size (as in *Figure 1I–L*). In this double-challenge condition, addition of 10 µM Dantrolene left the rapid induction portion of PHP intact (*Figure 6D–G*). Collectively, these experiments demonstrate that acute pharmacological perturbations targeting RyRs are capable of uncoupling the short-term induction and the long-term maintenance of PHP.

## Dual IP$_3$ sequestration and RyR blockade are not additive

In some tissues, RyR is activated by IP$_3$R-mediated Ca$^{2+}$ release, in a signaling process termed Calcium-Induced Calcium Release (CICR) (*Berridge, 1998*). IP$_3$Rs and RyRs have been placed together in CICR signaling processes in other systems, and our group has also identified overlapping functions of IP$_3$R and RyR at the NMJ (*Brusich et al., 2018*). Thus, we tested if IP$_3$ signaling and RyR functions might support the maintenance of PHP at the *Drosophila* NMJ via a shared process. The expectation for a shared process would be that a dual block of PHP (consisting of IP$_3$ sequestration +RyR pharmacological blockade) would not depress evoked transmission or quantal content below either individual manipulation.

A chronic *GluRIII[RNAi]*-expression NMJ challenge is amenable both to pharmacology and dual-tissue *UAS-IP$_3$-sponge* expression (*Pre-+Post* Gal4). In the *GluRIII[RNAi]* genetic background, both 10 µM Dantrolene application and *UAS-IP$_3$-sponge* expression blocked the long-term maintenance of PHP (*Figure 7A–C*). This resulted in EPSPs that were blunted compared to *GluRIII[RNAi]*-alone controls (*Figure 7B,D*). When we combined Dantrolene application and *UAS-IP$_3$-sponge* expression in the *GluRIII[RNAi]* background, the PHP impairment was indistinguishable from the impairment elicited by Dantrolene alone or *UAS-IP$_3$-sponge* alone (*Figure 7A–D*) (*Supplementary file 6* for summary *Figure 7* data) (*Figure 7—source data 1* for raw data). The data are consistent with a model in which IP$_3$ sequestration and RyR blockade disrupt the long-term maintenance of PHP as part of a shared process, either via a single linear pathway or convergent pathways.

## Axotomy does not block rapid PHP, even when IP$_3$ signaling is impaired

For larval NMJ electrophysiology, motor neurons are severed several minutes before recording (*Jan and Jan, 1976*). Although standard practice, this procedure requires special attention in our study because store-operated calcium release mediates a variety of cellular responses after axotomy in rodent (*Rigaud et al., 2009*) and nematode models (*Sun et al., 2014*). For the rapid induction of PHP at the NMJ, PhTox is typically applied to intact synapses, prior to motor nerve severing and recording (*Frank et al., 2006*). This allows for endogenous spontaneous activity to drive PhTox to bind to open channels prior to recording (*Frank et al., 2006*). Nevertheless, rapid induction of PHP

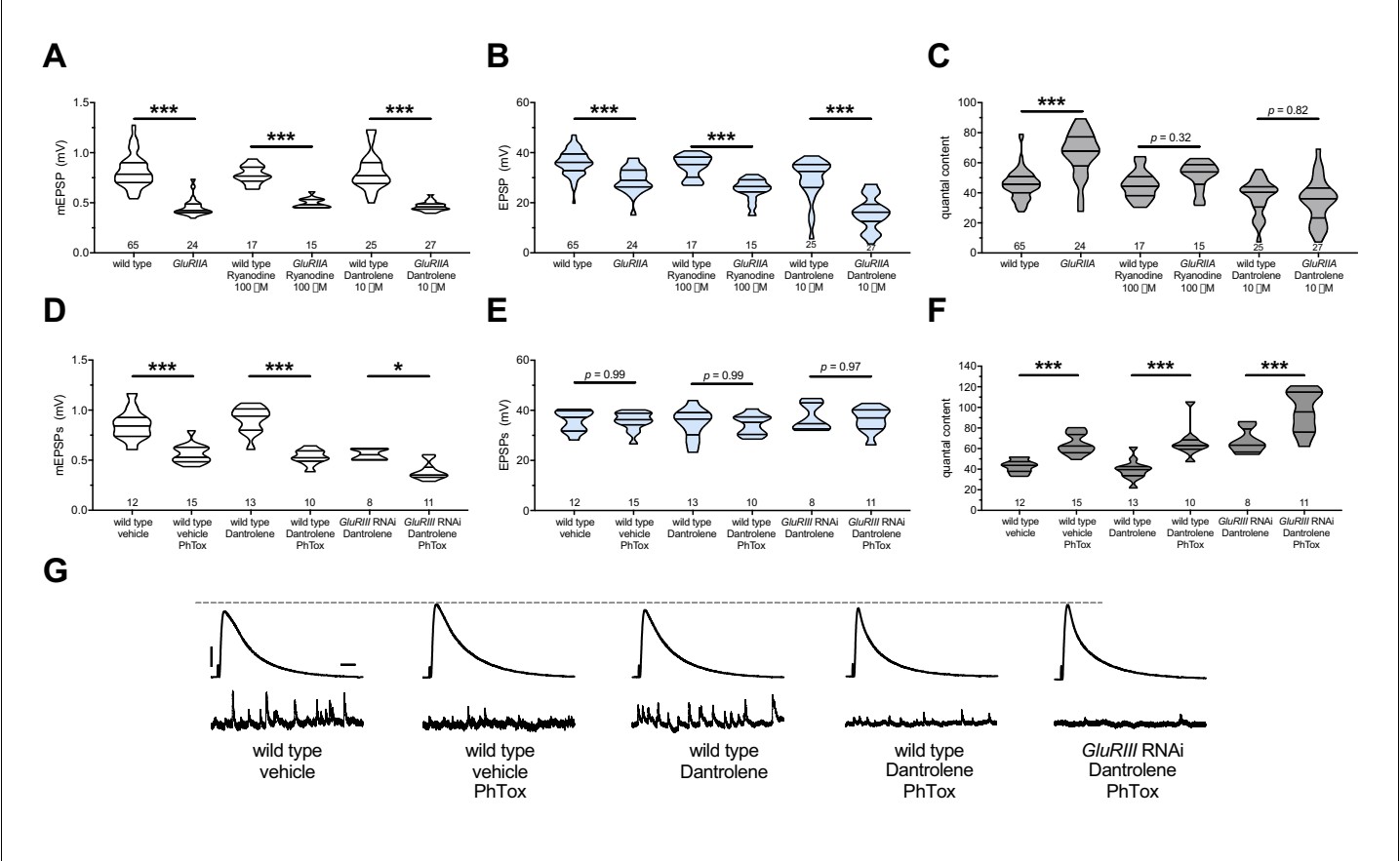

**Figure 6.** The maintenance of PHP requires continuous RyR function, but PHP induction does not. Ryanodine or Dantrolene acutely applied to NMJs to impair RyR function. Wild-type and *GluRIIA* control data sets are replotted from *Figures 4* and *5* for visual comparison. (A) The *GluRIIA^SP16* deletion mutation diminishes quantal size for all experimental conditions. (B) EPSP amplitudes are somewhat impaired versus non-*GluRIIA* control in all cases but most severely impaired when *GluRIIA* deletion is combined with 10 µM Dantrolene. (C) By quantal content, sustained PHP expression is abolished with acute NMJ exposure to 100 µM Ryanodine or 10 µM Dantrolene. (D) Acute incubation 20 µM PhTox diminishes quantal size for all conditions shown. (E) EPSP amplitudes remain near genetic control levels for all conditions with PhTox application. (F) Rapid PHP induction by PhTox is intact in the presence of 10 µM Dantrolene, even when *GluRIII* has been knocked down throughout life. (G) Representative electrophysiological traces for the conditions in (D–F). Violin plots used as in *Figure 1*. Statistical comparisons are by Student's T-Test vs. unchallenged controls. *p<0.05, **p<0.01, ***p<0.001. Scale bars for all traces are $y$ = 10 mV (1 mV), $x$ = 20 ms (500 ms) for EPSPs (mEPSPs).

DOI: https://doi.org/10.7554/eLife.39643.012

The following source data is available for figure 6:

**Source data 1.** Raw electrophysiology data for *Figure 6*.
DOI: https://doi.org/10.7554/eLife.39643.013

still works effectively when the motor nerves are severed prior to PhTox exposure (*Frank et al., 2006*).

To test for a possible synergistic interaction between axotomy, IP₃-directed signaling, and ER function during the acute induction phase of PHP, we applied PhTox to NMJs with intact motor nerves or with cut motor nerves (central nervous system (CNS) excised). We did this in a genetic background while expressing *UAS-IP₃-sponge* construct pre- and postsynaptically. For controls, we treated the preparations identically and used GAL4 drivers alone for the genetic background. We found that rapid PHP induction still worked in the *UAS-IP₃-sponge*-expressing background, regardless of whether the motor nerve was severed prior to PhTox application (*Figure 7E–G*). The evoked events were slightly diminished for the *UAS-IP₃-sponge* expressing NMJs where the CNS was cut out of the preparation prior to PhTox application (*Figure 7F,H*). However, by quantal content measures, the rapid induction of PHP was not blocked by this dual treatment (*Figure 7G*).

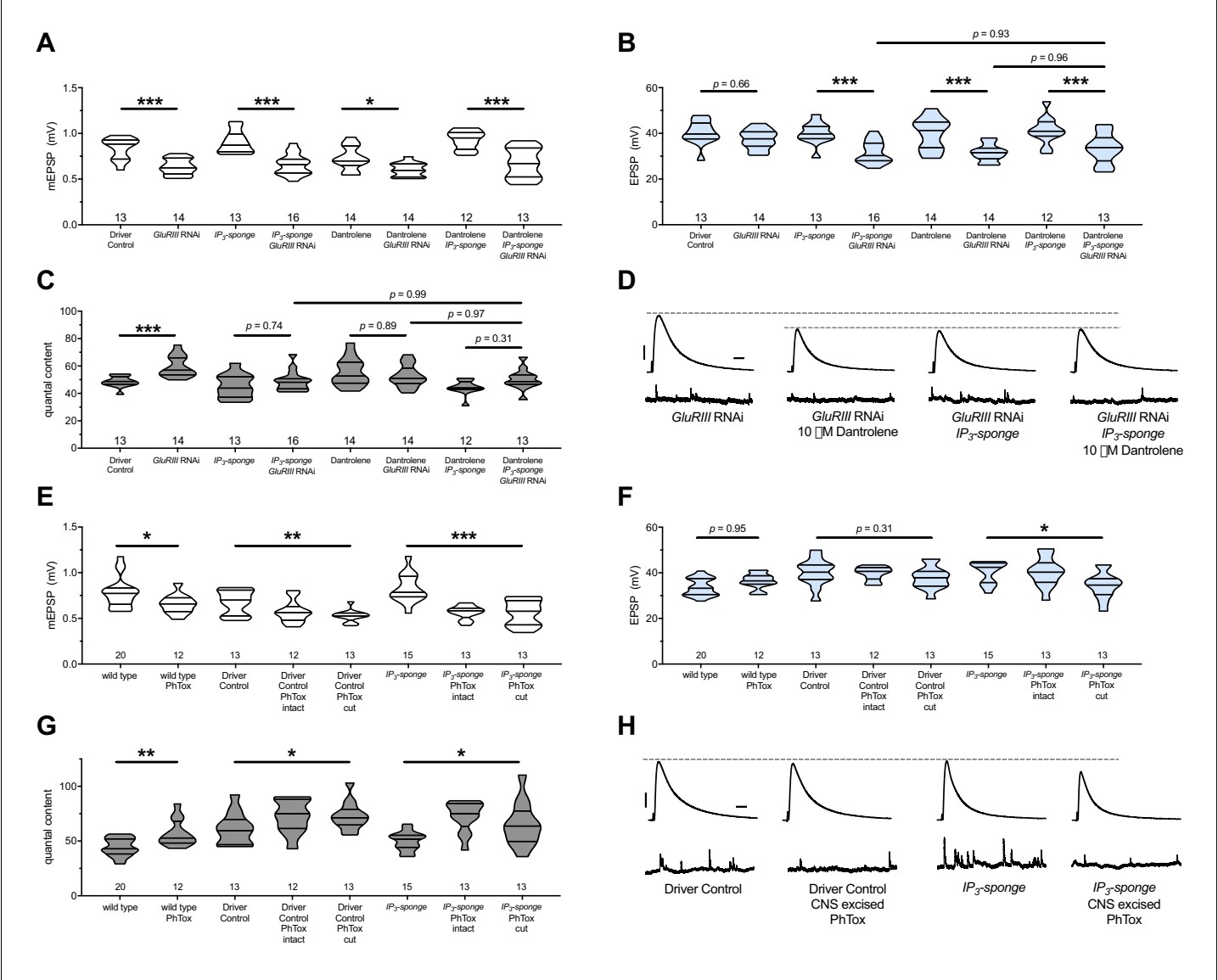

**Figure 7.** There are no additive effects of genetic IP$_3$ signaling inhibition and pharmacological RyR inhibition. (**A**) *GluRIII* knockdown diminishes quantal size for all experimental conditions. (**B**) When challenged with *GluRIII* knockdown, EPSP amplitudes are maintained for the GAL4 driver control background but impaired for all other experimental backgrounds. The dual manipulation of 10 μM Dantrolene +*UAS-IP$_3$-sponge* is indistinguishable from the single manipulations alone. (**C**) By quantal content, sustained PHP expression is abolished with chronic IP$_3$ sequestration, acute 10 μM Dantrolene application, or both. (**D**) Representative electrophysiological traces for the conditions in (**A–C**). (**E**) Acute incubation 20 μM PhTox diminishes quantal size for all conditions shown. (**F**) EPSP amplitudes remain near genetic control levels for all conditions with PhTox application. There is a slight diminishment for the condition in which *UAS-IP$_3$-sponge* is expressed and PhTox application is performed only after nerve dissection. (**G**) Full, rapid PHP induction or partial PHP induction by PhTox is present for all conditions shown. (**H**) Representative electrophysiological traces for the conditions in (**E–G**). Violin plots used as in *Figure 1*. Statistical comparisons for (**A–C**) and for wild type vs. wild type +PhTox are by Student's T-Test vs. unchallenged controls. Statistical comparisons across three data sets are by one-way ANOVA followed by Bonferroni post-hoc test across genotypes shown. *p<0.05, **p<0.01, ***p<0.001. Scale bars for all traces are *y* = 10 mV (1 mV), *x* = 20 ms (500 ms) for EPSPs (mEPSPs).

DOI: https://doi.org/10.7554/eLife.39643.014

The following source data is available for figure 7:

**Source data 1.** Raw electrophysiology data for *Figure 7*.

DOI: https://doi.org/10.7554/eLife.39643.015

## Neuron and muscle IP$_3$ signaling both contribute to long-term homeostatic potentiation

Insofar, none of the genetic or pharmacological manipulations impairing PHP maintenance in this study have been tissue specific. In principle, all PHP-blocking manipulations described could operate either in neuronal or muscle substrates – or upon both tissues. Our prior work showed that chronic *Plc21C* gene knockdown in the muscle alone is not sufficient to impair PHP (*Brusich et al., 2015*). That result suggested a neuronal component to this signaling system for PHP maintenance. Yet further tests are needed. We wished to understand whether a pre- or postsynaptic mechanism (or a dual-tissue mechanism) governs IP$_3$-mediated Ca$^{2+}$ store release signaling in order to support long-term maintenance of PHP.

We turned again to the *UAS-IP$_3$-sponge* transgene because it can be expressed in a tissue-specific manner, and it conveyed a full block of PHP when dually expressed in the neuron and muscle (*Figure 2*). We expressed *UAS-IP$_3$-sponge* in a GAL4 driver (neuron or muscle alone) or a driver +*GluRIIASP$^{SP16}$* null genetic background. We then quantified PHP by NMJ electrophysiology, considering neuronal (*Figure 8A–C*) or muscle (*Figure 8D–F*) expression (*Supplementary file 7* for summary *Figure 8* data) (*Figure 8—source data 1* for raw data). Surprisingly, we did not localize the full block of PHP maintenance to a single tissue. For expression in either tissue alone, there was still a small increase in QC in a *Gal4 >>UAS-IP$_3$-sponge + GluRIIASP$^{SP16}$* genetic condition, compared to control *Gal4 >>UAS-IP$_3$-sponge* expression in an unchallenged background (*Figure 8C,F*). EPSP values were depressed when *UAS-IP$_3$-sponge* was expressed neuronally in the *GluRIIA* null background (*Figure 8B,G*), consistent with an important neuronal component to PHP (*Brusich et al., 2015*). Combined with our prior data (*Figure 2*), we conclude that the maintenance of PHP can be fully erased by IP$_3$ sequestration – but only if this is done in a dual tissue manner.

Our data indicate that IP$_3$ functions in a shared process with Ca$^{2+}$ store release. Presynaptic neurotransmitter release at the NMJ and other synapses is highly sensitive to changes in intracellular Ca$^{2+}$ concentration after influx through voltage-gated Ca$_V$2 channels. Therefore, we checked if IP$_3$ signaling and its effects on intracellular Ca$^{2+}$ release might impinge upon the Ca$^{2+}$ sensing machinery in the presynaptic cleft, which could potentially influence PHP. We conducted NMJ recordings over a range of low extracellular [Ca$^{2+}$] and calculated the Ca$^{2+}$ cooperativity of release for the dual-tissue expression *UAS-IP$_3$-sponge* NMJs, as well as wild-type NMJs and GAL4 driver control NMJs. The Ca$^{2+}$ cooperativity of release was steady between the three conditions (*Figure 8H,I*), indicating that sequestration of cellular IP$_3$ at the NMJ does not directly alter the Ca$^{2+}$-dependence of synaptic release at the presynaptic NMJ.

## Discussion

In this study, we divided the acute induction and chronic maintenance stages of presynaptic homeostatic potentiation. Our data support two core findings. The first is that the short-term induction and long-term maintenance of PHP are separable by genetic and pharmacological manipulations. The second is that an IP$_3$-mediated signaling system is specifically required for the maintenance of PHP (*Figure 9*).

### Acute versus chronic PHP expression

For several years, one assumption has been that both the acute and chronic forms of PHP are executed in a similar way – and possibly by shared mechanisms. The issue has been clouded by the fact that both PhTox and a *GluRIIA* deletion mutant – the primary reagents utilized to induce PHP – have the same molecular target, that is GluRIIA-containing glutamate receptors (*Frank et al., 2006*; *Petersen et al., 1997*). The process of combining these acute and chronic forms of plasticity within a single genotypic background was cumbersome due to a lack of reagents available to conduct temporally separate targeting experiments.

Several groups ascertained insights into temporal requirements by targeting potential homeostatic signaling genes. The main finding has been that the majority of molecules identified are essential to both the acute and chronic forms of PHP (*Davis and Müller, 2015*; *Frank, 2014a*). Neurons tightly control neurotransmitter release probability, and the core presynaptic machinery directly responsible for increasing quantal content is shared. These shared components include the Ca$_V$2-

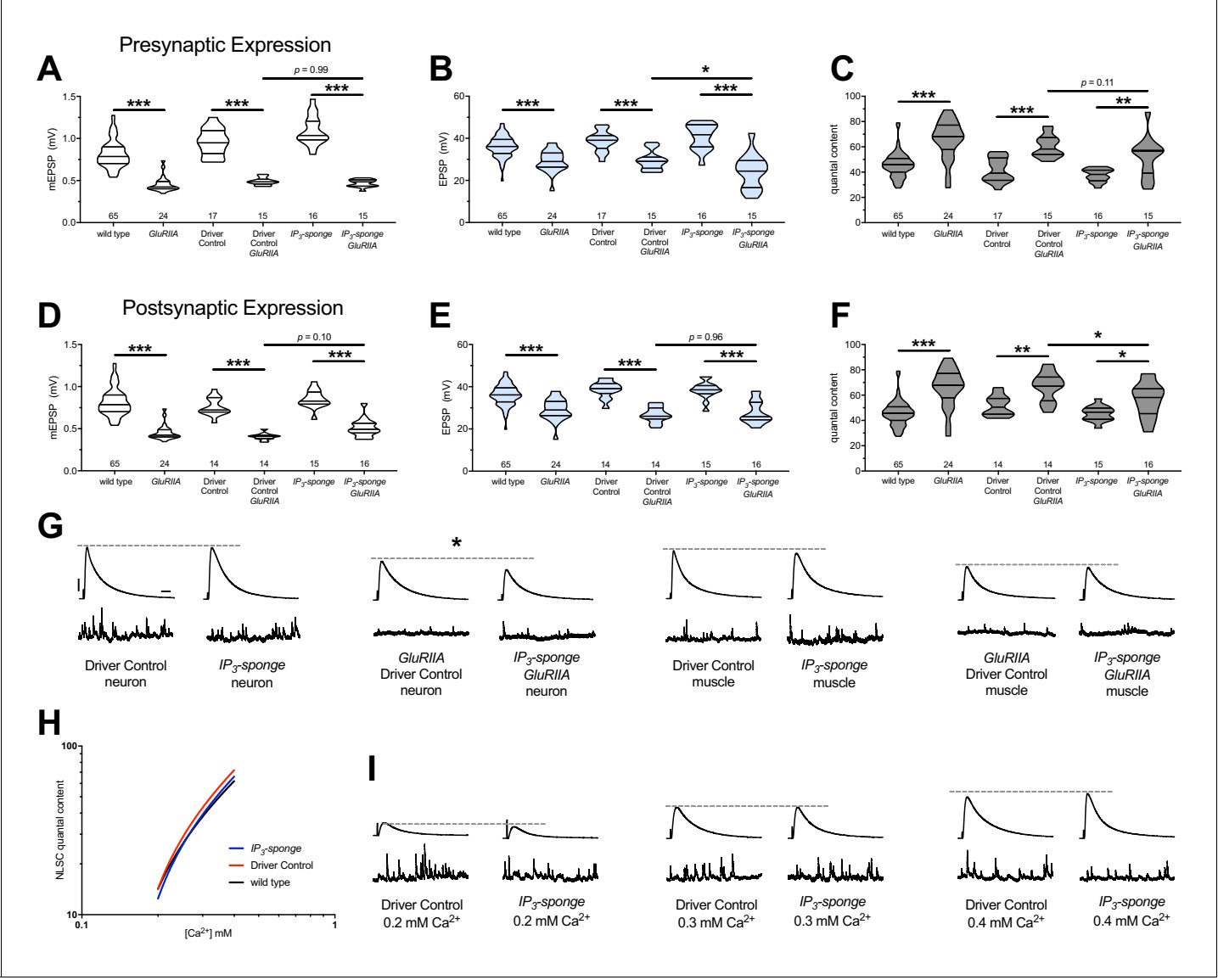

**Figure 8.** Combined pre-and postsynaptic IP₃ signaling maintains PHP. *UAS-IP₃-sponge* transgene expression in single tissue types impairs PHP maintenance, but does not block it. *IP₃-sponge* either in neurons (A–C) or muscle (D–F). Wild-type and *GluRIIA* control data sets are replotted from *Figures 4–6* for visual comparison. (A) The *GluRIIA^SP16^* deletion mutation diminishes quantal size for all experimental conditions. (B) EPSP amplitudes are somewhat impaired versus non-*GluRIIA* control in all cases but most severely impaired when *GluRIIA* deletion is combined with presynaptic *IP₃-sponge* expression. (C) By quantal content, sustained PHP is still present for all conditions shown. (D) The *GluRIIA^SP16^* deletion mutation diminishes quantal size for all experimental conditions. (E) EPSP amplitudes are somewhat impaired versus non-*GluRIIA* control. (F) By quantal content, sustained PHP is still present for all conditions shown. (G) Representative electrophysiological traces for conditions in (A–F). (H) *UAS-IP₃-sponge* transgene expression does not impair calcium cooperativity of release. (I) Representative electrophysiological trances for conditions in (H). Violin plots used as in *Figure 1*. Statistical comparisons are by one-way ANOVA followed by Bonferroni post-hoc test across genotypes shown. *p<0.05, **p<0.01, ***p<0.001. Scale bars in (G) apply to all traces in (G) and (I) and are *y* = 10 mV (1 mV), *x* = 20 ms (500 ms) for EPSPs (mEPSPs).
DOI: https://doi.org/10.7554/eLife.39643.016

The following source data is available for figure 8:

**Source data 1.** Raw electrophysiology data for *Figure 8*.
DOI: https://doi.org/10.7554/eLife.39643.017

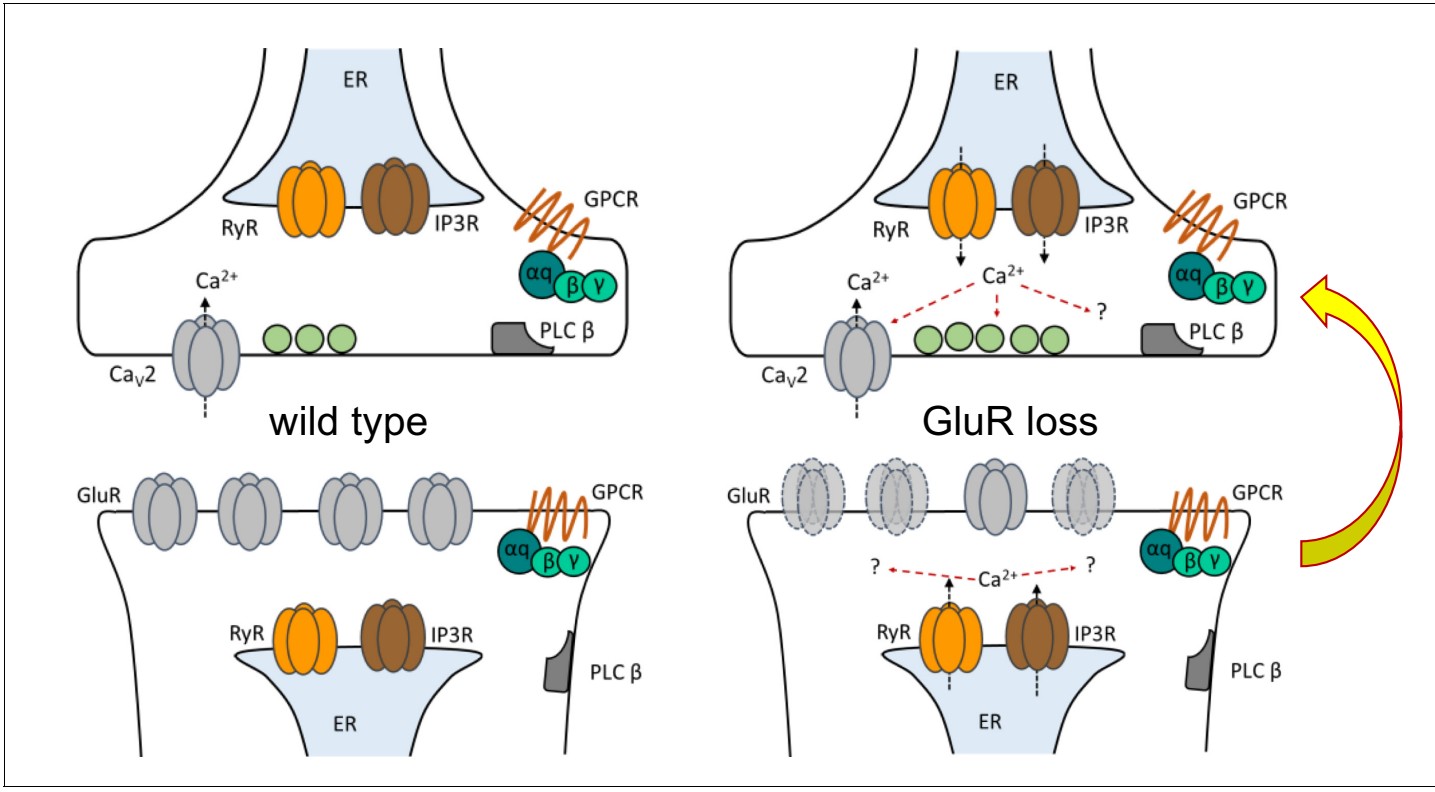

**Figure 9.** Model depicting PLCβ/IP3R/RyR signaling underling the maintenance of PHP in both muscle and neuron. At the *Drosophila* NMJ, PLCβ and effectors IP₃R and RyR are required for the maintenance of HSP. Left: PLCβ signaling components depicted in both muscle and neuron at the *Drosophila* NMJ. We detected no apparent role for PLCβ, IP₃R, or RyR in baseline neurotransmission. Right: Reduced postsynaptic glutamate receptor function – either due to deletion of the *GluRIIA* gene or expression of *UAS-GluRIII[RNAi]* – drives a chronic form of PHP that is maintained throughout life. A retrograde, muscle-to-nerve signal instructs the neuron to increase the number of neurotransmitter vesicles released (quantal content). Our data support a model in which long-term maintenance of PHP requires PLCβ and its effectors in both the presynaptic neuron and postsynaptic muscle, but these factors are dispensable for the rapid induction of PHP.

DOI: https://doi.org/10.7554/eLife.39643.018

type voltage-gated calcium channel or factors gating influx through the channel (*Frank et al., 2006*; *Frank et al., 2009*; *Müller and Davis, 2012*; *Wang et al., 2014*; *Wang et al., 2016a*; *Younger et al., 2013*). They also include factors that regulate the size of the readily releasable pool (RRP) of presynaptic vesicles (*Harris et al., 2015*; *Müller et al., 2015*; *Müller et al., 2012*; *Wang et al., 2016a*; *Wentzel et al., 2018*; *Weyhersmüller et al., 2011*) or factors that control the baseline excitability or plasticity of the presynaptic motor neuron (*Bergquist et al., 2010*; *Kiragasi et al., 2017*; *Orr et al., 2017b*; *Parrish et al., 2014*; *Younger et al., 2013*) – and neurotransmitter fusion events themselves (*Dickman and Davis, 2009*; *Dickman et al., 2012*; *Müller et al., 2011*; *Ortega et al., 2018*). As a result, both the acute and chronic forms of PHP signaling cause increases in readily releasable pool (RRP) size and Ca$_V$2-mediated calcium influx; and in turn, these presynaptic mechanisms underlie the increases in QC which constitute PHP (*Davis and Müller, 2015*; *Müller and Davis, 2012*; *Müller et al., 2012*).

Here, we show that although the acute and chronic processes might overlap, they are functionally separable. The fact that they are separable is not necessarily surprising. This finding mirrors data for discrete molecules required for long-term PHP maintenance, such as Target of Rapamycin (Tor) (*Goel et al., 2017*; *Kauwe et al., 2016*; *Penney et al., 2012*), the Rho-type guanine exchange factor Ephexin (*Frank et al., 2009*), the transcription factor Gooseberry (*Marie et al., 2010*), C-terminal Src Kinase (*Spring et al., 2016*), innate immune signals molecules IMD, IKKβ, Relish (*Harris et al., 2018*) and the kinesin adaptor Arl8 (*Goel et al., 2019a*; *Vukoja et al., 2018*). Importantly, this list contains molecules implicated both in neuron and muscle. We have added PLCβ (*Brusich et al., 2015*) and its effectors IP₃R and RyR to this list.

Recent studies have augmented the idea of overlapping signaling pathways and added a degree of specificity. Both acute and chronic forms of PHP begin as instructive retrograde signals after perturbations are detected in the muscle (*Hauswirth et al., 2018*; *Orr et al., 2017a*). These forms of PHP involve a decrease in phosphorylation of muscle CaMKII levels, and converge upon the same signaling components in the presynaptic neuron (*Goel et al., 2017*; *Li et al., 2018*; *Newman et al., 2017*). These studies suggest that Tor signaling converges on the same molecular targets as acute forms of PHP (*Goel et al., 2017*). However, the precise roles for Tor and CaMKII in either form of PHP are as yet unknown.

Our data appear to contradict the idea of PHP pathway convergence (*Goel et al., 2017*). Yet, our findings are not incompatible with this idea. Multiple lines of evidence indicate discrete signaling requirements for acute forms of PHP on both sides of the synapse. A convergence point is undefined. Accounting for the separation of acute and chronic forms of PHP – as well as their discrete signaling requirements – long-term maintenance of PHP might integrate multiple signals between the muscle and neuron over time. For future studies, it will be important to clearly define roles of signaling systems underlying PHP and how distinct signaling systems might be linked.

## Unexpected findings about PHP stage separation

Our work presents unexpected findings. The first is that even in the face of a chronic impairment or block of homeostatic potentiation, the NMJ is nevertheless capable of a full rapid induction of PHP (*Figures 1*, *2* and *7*). Given that most molecules required for PHP identified to date are needed for both phases, we did not expect significant functional separation between them. We expected a priori that a failure of the chronic maintenance of PHP would make core machinery unavailable for its acute induction. The second unexpected finding is how quickly the chronic maintenance of PHP can be nullified by pharmacology (10 min), resulting in a return to baseline neurotransmitter release probability after only minutes of drug exposure (*Figures 4–6*). We showed that homeostatic potentiation in *GluRIIA* mutant larvae or *GluRIII* knock-down larvae was abrogated by four different reagents previously known to block $IP_3R$ (*Figures 4* and *5*) or RyR (*Figure 6*). Those findings are reminiscent of prior work showing that acute blockade of DAG/ENaC channels with the drug benzamil abolishes PHP in both a *GluRIIA* mutant background, as well as in the presence of PhTox (*Younger et al., 2013*). A difference between benzamil application and the pharmacological agents used in our study is that the drugs we employed only abolished PHP in a chronically challenged background.

## Does PHP induction lead to maintenance?

It is unclear how signaling systems that drive homeostatic plasticity transition from a state of induction to a state of maintenance. It is also not understood how interdependent short-term and long-term HSP implementation mechanisms are. A more complete understanding of the timing and perdurance of these properties could have important implications for neurological conditions where synapse stability is episodically lost (*Russell et al., 2013*).

Our findings parallel recent data examining active zone protein intensities in the contexts of induction of PHP and maintenance of PHP at the NMJ. There are multiple results informative to our study. First, the expression of any form of PHP (acute or chronic) appears to correlate with an increased intensity of active zone protein levels, such as $Ca_V2$/Cacophony (*Gratz et al., 2019*), UNC-13 (*Böhme et al., 2019*), and the Drosophila CAST/ELKS homolog Bruchpilot (*Böhme et al., 2019*; *Goel et al., 2019a*; *Gratz et al., 2019*). Unexpectedly, however, two of these studies also reported that the rapid induction of PHP does not require this protein increase in order to be functionally executed (*Böhme et al., 2019*; *Goel et al., 2019a*). These are conundrums for future work. How does rapid active zone remodeling happen in minutes on a mechanistic level? In the absence of such remodeling, how is PHP able to be induced rapidly? Moreover, is the observed short-term active zone remodeling the kernel for the longer-term changes to the active zone and release probability – or is some other compensatory system triggered over long periods of developmental time (e.g., see multiple mechanisms described by *Goel et al., 2019b*)?

Our findings add a new dimension to those puzzles with the data that $IP_3$ signaling is continuously required to maintain PHP. If active zone remodeling truly is instructive for PHP maintenance, then it will be interesting to test what roles $IP_3$ signaling and intracellular calcium release play in that

process. Our screen did include a *UAS-RNAi* line against *unc-13* and an upstream GPCR-encoding gene *methuselah* (*Supplementary file 2*). Moreover, we previously published a study of PHP using a *UAS-cac[RNAi]* line (*Brusich et al., 2015*). Chronic PHP maintenance was intact for all of those manipulations. Those findings are not necessarily contradictory to the recent work from other groups. For instance, knockdown of an active zone protein by RNAi is not a null condition. As such, RNAi-mediated knockdown should leave residual wild-type protein around. In theory, that residual protein could be scaled with homeostatic need.

## PLCβ- and IP$_3$-directed Signaling is Required for PHP Maintenance

Our data strongly suggest that intracellular calcium channel activation and store release fine tune neurotransmitter release that is implemented by PHP. The exact mechanism by which IP$_3$R and RyR function to maintain PHP at the NMJ is unclear. It appears to be a shared process with IP$_3$ (*Figure 7*). If these store-release channels are acting downstream of IP$_3$ activity, then our data suggest that this would be a coordinated activity involving both the muscle and the neuron (*Figures 8* and *9*) – with loss of IP$_3$ signaling in the neuron being more detrimental to evoked release (*Figure 8G*).

It remains unclear what signals are acting upstream. PLCβ is canonically activated by Gαq signaling. From our prior work, we garnered evidence that a *Drosophila* Gq protein plays a role in the long-term maintenance of HSP (*Brusich et al., 2015*). Logically, there may exist a G-protein-coupled receptor (GPCR) that functions upstream of PLCβ/IP$_3$ signaling. Our screen did not positively identify such a GPCR. We did examine several genes encoding GPCRs, including *TkR86C*, *mAChR-A*, *GABA-B-R1*, *PK2-R2*, *methuselah*, *AdoR*, and *mGluR* (*Supplementary file 2*). We also examined genes encoding Gβ subunits or putative scaffolding molecules, including *CG7611* (a WD40-repeat-encoding gene), *Gβ13F*, and *Gβ76C*, again with no positive screen hits (*Supplementary file 2*).

Our data are consistent with dual pre- and postsynaptic functions of IP$_3$. This could mean dual pre- and postsynaptic roles for calcium store release – through an undetermined combination of RyR and IP$_3$R activities, again either pre- or postsynaptic. Both RyR and IP$_3$R have been shown to be critical for specific aspects of neuroplasticity and neurotransmission (*Berridge, 2016*). Activities of both RyR and IP$_3$R can activate molecules that drive plasticity, such as Calcineurin (*Victor et al., 1995*) and CaMKII (*Shakiryanova et al., 2011*). At rodent hippocampal synapses, electrophysiological measures like paired-pulse facilitation and frequency of spontaneous neurotransmitter release (*Emptage et al., 2001*) are modulated by RyR and/or IP$_3$R function, as is facilitation of evoked neurotransmitter at the rat neocortex (*Mathew and Hablitz, 2008*). In addition to vesicle fusion apparatus, activity of presynaptic voltage-gated calcium channels is modulated by intracellular calcium (*Catterall, 2011*; *Lee et al., 2000*). Our own work at the NMJ has shown that impairing factors needed for store-operated calcium release can mollify hyperexcitability phenotypes caused by gain-of-function Ca$_V$2 amino-acid substitutions (*Brusich et al., 2018*).

Within the presynaptic neuron, IP$_3$R and RyR could activate any number of calcium-dependent molecules to propagate homeostatic signaling. We tested some candidates in our screen (*Figure 2*, *Supplementary file 2*), but none of those tests blocked PHP. One possibility is that the reagents we utilized did not sufficiently diminish the function of target molecules enough to impact PHP in this directed screen. Detection of downstream effectors specific to muscle or neuron might also be hampered by the fact that attenuation of IP$_3$ signaling in a single tissue is insufficient to abrogate PHP. Another possibility is that presynaptic store calcium efflux via IP$_3$R and RyR may directly potentiate neurotransmitter release, either by potentiating basal calcium levels or synchronously with Ca$_V$2-type voltage-gated calcium channels (*Frank et al., 2006*; *Müller and Davis, 2012*).

Both pre- and postsynaptic voltage-gated calcium channels are critical for the expression of several forms of homeostatic synaptic plasticity (*Frank, 2014b*). Much evidence supports the hypothesis that store-operated channels and voltage gated calcium channels interact to facilitate PHP. In various neuronal populations, both RyR and IP$_3$R interact with L-type calcium channels physically and functionally to reciprocally impact the opening of the other channel (*Chavis et al., 1996*; *Kim et al., 2007*; *Ouardouz et al., 2003*). In presynaptic boutons, RyR calcium release follows action potential firing (*Emptage et al., 2001*). Calcium imaging experiments show that both the acute expression and sustained maintenance of PHP requires an increase in presynaptic calcium following an action potential (*Müller and Davis, 2012*). Because IP$_3$Rs are activated by both free calcium and IP$_3$, elevated IP$_3$ levels in the case of chronically expressed PHP could allow IP$_3$Rs and RyRs to open in a

way that is time-locked with Ca$_V$2-mediated calcium influx or in a way to facilitate the results of later Ca$_V$2-mediated influx.

# Materials and methods

## Key resources table

| Reagent type (species) or resource | Designation | Source or reference | Identifiers | Additional information |
|---|---|---|---|---|
| Genetic Reagent (*Drosophila melanogaster*) | *GluRIII[RNAi]* or *UAS-GluRIII[RNAi]* | PMID: 25859184 | FlyBase ID: FlyBase_FBtp0110520 | *UAS-pWiz* transgene knocking down *GluRIII* gene function. This lab (CAF) is the source (*Brusich et al., 2015*). |
| Genetic Reagent (*D. melanogaster*) | *Plc21C[RNAi]* or *UAS-Plc21C[RNAi]* | Vienna Drosophila Resource Center (GD11359); PMID: 17625558 | RRID:FlyBase_FBst0456476 | *UAS-RNAi* transgene |
| Genetic Reagent (*D. melanogaster*) | *Plc21C[RNAi]* or *UAS-Plc21C[RNAi]* | Vienna Drosophila Resource Center (GD11359); PMID: 17625558 | RRID:FlyBase_FBst0456477 | *UAS-RNAi* transgene |
| Genetic Reagent (*D. melanogaster*) | *UAS-IP$_3$-sponge.m30* | PMID: 16540404 | FlyBase ID: FlyBase_FBtp0068098 | also referred to as *UAS-IP$_3$-sponge* |
| Genetic Reagent (*D. melanogaster*) | *UAS-IP$_3$-sponge.m49* | PMID: 16540404 | FlyBase ID: FlyBase_FBtp0068099 | also referred to as *UAS-IP$_3$-sponge* |
| Genetic Reagent (*D. melanogaster*) | *GluRIIA$^{SP16}$* | PMID: 9427247 | RRID:BDSC_64202 | deletion allele; also referred to as *GluRIIA* |
| Genetic Reagent (*D. melanogaster*) | *w$^{1118}$* | PMID: 6319027 | RRID:BDSC_3605 | wild-type genetic background |
| Genetic Reagent (*D. melanogaster*) | *elaV(C155)-Gal4* | PMID: 7917288 | RRID:BDSC_458 | also known as *C155-Gal4* |
| Genetic Reagent (*D. melanogaster*) | *Sca-Gal4* | PMID: 8893021 | FlyBase ID: FlyBase_FBtp0007534 | |
| Genetic Reagent (*D. melanogaster*) | *BG57-Gal4* | PMID: 8893021 | FlyBase ID: FlyBase_FBti0016293 | also known as *C57-Gal4* |
| Chemical Compound, Drug | Philanthotoxin-433; PhTox | Sigma-Aldrich (MilliporeSigma); Santa Cruz Biotechnology | CAS Number: (Sigma-Aldrich and Santa Cruz Biotechnology)_276684-27-6 | product P207 discontinued by Sigma-Aldrich |
| Chemical Compound, Drug | Xestospongin C | Abcam | CAS Number: Abcam_88903-69-9 | |
| Chemical Compound, Drug | 2-APB | Tocris | CAS Number: Tocris_524-95-8 | |
| Chemical Compound, Drug | Ryanodine | Tocris | CAS Number: Tocris_15662-33-6 | |

*Continued on next page*

*Continued*

| Reagent type (species) or resource | Designation | Source or reference | Identifiers | Additional information |
|---|---|---|---|---|
| Chemical Compound, Drug | Dantrolene | Tocris | CAS Number: Tocris_14663-23-1 | |
| Antibody | Monoclonal mouse anti-Synapsin | DSHB (3C11) | Cat#: DSHB_3C11; RRID: AB_2313867 | (1:50) |
| Antibody | Polyclonal rabbit anti-Dlg | PMID: 8893021 | | (1:15,000) |
| Antibody | Polyclonal goat anti-mouse 488 (DyLight) | Jackson ImmunoResearch | Cat #:Jackson_115-485-003; (no RRID) | (1:1000) discontinued; substitute with Cat# 115-485-068; RRID:AB_2338804 |
| Antibody | Polyclonal goat anti-rabbit 549 (DyLight) | Jackson ImmunoResearch | Cat#:Jackson_111-505-003; RRID:AB_2493180 | (1:2000) discontinued; substitute with Cat# 111-165-003; RRID:AB_2338000 |
| Antibody | Polyclonal goat anti-HRP (Alexa-647) | Jackson ImmunoResearch | Cat#:Jackson_123-605-021; RRID:AB_2338967 | (1:250) |
| Software, Algorithm | pClamp | Molecular Devices | RRID:SCR_011323 | |
| Software, Algorithm | MiniAnalysis Program | Synaptosoft | RRID:SCR_002184 | |
| Software, Algorithm | GraphPad Prism | GraphPad | RRID:SCR_002798 | |

## *Drosophila* husbandry

*Drosophila melanogaster* fruit flies were raised on Cornmeal, Molasses and Yeast Medium prepared according to the Bloomington Drosophila Stock Center (BDSC, Bloomington, IN) recipe. *Drosophila* husbandry was performed according to standard practices (*Greenspan, 2004*). Larvae were raised at 25°C or 29°C in humidity controlled and light-controlled Percival DR-36VL incubators (Geneva Scientific).

## *Drosophila* genetic lines

$w^{1118}$ (*Hazelrigg et al., 1984*) was used as a non-transgenic wild type stock. The deletion *GluRIIA* allele (*GluRIIA$^{SP16}$*) was generated previously (*Petersen et al., 1997*). *UAS-IP$_3$-sponge* lines (*UAS-IP$_3$-sponge.m30* and *UAS-IP$_3$-sponge.m49*) were provided by Drs. Masayuki Koganezawa and Dai-suke Yamamoto (*Usui-Aoki et al., 2005*). The *UAS-GluRIII[RNAi]* line utilized to screen homeostatic candidate molecules was described previously (*Brusich et al., 2015*). GAL4 drivers simultaneously utilized for the 'Pre-+Post-Gal4' conditions were *elaV(C155)-Gal4* (*Lin and Goodman, 1994*), *Sca-Gal4* (*Budnik et al., 1996*), and *BG57-Gal4* (*Budnik et al., 1996*).

In addition to the *UAS-IP$_3$-sponge* lines, several *UAS-RNAi* or genetic mutant lines were obtained either from the BDSC or the Vienna Drosophila Resource Center (VDRC, Vienna, Austria). Those specific mutations and lines are detailed in *Supplementary file 2*. Procedures for how the *UAS-RNAi* lines were generated have been published (*Dietzl et al., 2007*; *Ni et al., 2009*).

## Electrophysiology and pharmacology

Wandering third instar larvae were collected and filleted for NMJ analysis. Control and experimental samples were collected in parallel, using identical conditions. Activity in abdominal muscle 6 from segments 2 and 3 was recorded in a modified HL3 saline (70 mM NaCl, 5 mM KCl, 5 mM HEPES, 10 mM NaHCO3, 115 mM sucrose, 0.5 mM CaCl$_2$ (unless otherwise noted), 10 mM MgCl2, 4.2 mM

trehalose, pH 7.2) (see *Stewart et al., 1994* for original parameters). Sharp electrode recordings of miniature excitatory postsynaptic potentials (mEPSPs) and excitatory postsynaptic potentials (EPSPs) were conducted as previously described (*Brusich et al., 2015*; *Spring et al., 2016*; *Yeates et al., 2017*) and analyzed using MiniAnalysis (Synaptosoft) and pClamp10 (Molecular Devices) software, blind to genotype or treatment. Uncorrected quantal content (QC) was estimated per NMJ as average EPSP/average mEPSP and was also reported as corrected for non-linear summation as done previously (*Martin, 1955*). For the correction factor formula (*Martin, 1955*), we used a reversal potential of +10 mV (Supplemental Excel File).

Pharmacological agents were bath applied in recording saline at the final concentrations indicated in the text, figures, and tables. The agents included Philanthotoxin-433 (PhTox, Sigma-Aldrich and Santa Cruz Biotechnology), Xestospongin C (Abcam), 2-APB (Tocris, Bio-Techne Corporation), Ryanodine (Tocris), and Dantrolene (Tocris).

To render mEPSP and EPSP traces for figures, we pulled (x,y) coordinates from the Clampfit program (Molecular Devices) and imported those coordinates into GraphPad Prism (GraphPad) software. For all traces, we chose a recording that was at (or closest to) the calculated average. For mEPSPs, we picked a representative selection of minis. For EPSPs, the final trace that was rendered was an average of all the EPSP traces from that particular NMJ.

### Immunostaining and analyses

Immunostaining and image analyses of NMJ 6/7 in segments A2 and A3 were performed as previously described (*Spring et al., 2016*; *Yeates et al., 2017*). Briefly, fileted larvae were fixed in Bouin's fixative for 4 min, washed, incubated in primary antibodies for 2 hr, washed, and incubated in secondary antibodies for an additional 2 hr. Bouton staining was performed to assess NMJ growth by using the following primary antibodies: mouse anti-Synapsin (anti-Syn; 3C11) 1:50 (Developmental Studies Hybridoma Bank, Iowa City, IA) and rabbit anti-Dlg 1:15,000 (*Budnik et al., 1996*). The following fluorophore conjugated secondary antibodies were also used (Jackson ImmunoResearch Laboratories): goat anti-mouse 488 1:1000 (DyLight) and goat anti–rabbit 549 1:2000 (DyLight). Goat anti-HRP 1:250 (Alexa-647) was utilized to stain neuronal membranes. Larval preparations were mounted in Vectashield (Vector Laboratories) and imaged at room temperature using Zen software on a Zeiss 880 Laser Scanning Microscope with an EC Plan-Neofluar 40X Oil DIC Objective (aperture 1.30) or an EC Plan-Apochromat 63X Oil DIC Objective (aperture 1.40; Zeiss). Experimental and control larval preps were imaged using identical acquisition settings and analyzed blind to genotype using the same procedure and thresholds. Images were prepared for publication in Adobe Photoshop using identical procedures for experimental and control images. Anti-Dlg bouton counts were completed in a blinded fashion to quantify synapse growth. For each anti-Dlg–positive bouton counted in muscle, it was verified that there was a corresponding cluster of anti-Syn staining in neurons.

### Statistical analyses

For electrophysiological data, statistical significance was tested either by Student's T-Test if one experimental data set was being directly compared to a control data set, or by one-way ANOVA with a Bonferroni post-hoc test if multiple data sets were being compared. For bouton counting, significance was tested utilizing a Kruskal-Wallis ANOVA test followed by Dunn's post-hoc test. Specific p value ranges and tests are noted in the Figures and Figure Legends and Supplementary Files and shown in graphs as follows: $*p < 0.05$, $**p < 0.01$, and $***p < 0.001$. All statistical analyses were conducted using GraphPad Prism Software. Most figure data are plotted as violin plots from GraphPad Prism; the violin plot shapes signify data distribution, $n$ values are below those shapes, and horizontal lines signify the $0^{th}$, $25^{th}$, $50^{th}$, $75^{th}$, and $100^{th}$ percentiles of the data.

## Acknowledgements

We thank members of the Frank lab for helpful comments on earlier versions of this study, and the laboratories of Drs. Tina Tootle, Fang Lin, Toshihiro Kitamoto, Pamela Geyer, and Lori Wallrath for helpful discussions. We also thank Drs. Andrew Russo, N Charles Harata, and Christopher Ahern for helpful suggestions. We thank the Bloomington Drosophila Stock Center and the Vienna Drosophila

Resource Center for several fly stocks detailed in the Materials and methods Section and Supplementary Files.

# Additional information

## Funding

| Funder | Grant reference number | Author |
| --- | --- | --- |
| Whitehall Foundation | 2014-08-03 | C Andrew Frank |
| National Science Foundation | 1557792 | C Andrew Frank |
| National Institute of Neurological Disorders and Stroke | R01NS085164 | C Andrew Frank |
| National Institute of Neurological Disorders and Stroke | T32NS007421 - PI Daniel T Tranel | Thomas D James |

The funders had no role in study design, data collection and interpretation, or the decision to submit the work for publication.

## Author contributions

Thomas D James, Conceptualization, Data curation, Formal analysis, Investigation, Methodology, Writing—original draft, Writing—review and editing; Danielle J Zwiefelhofer, Formal analysis, Investigation, Methodology, Writing—review and editing; C Andrew Frank, Conceptualization, Data curation, Formal analysis, Supervision, Funding acquisition, Methodology, Writing—original draft, Project administration, Writing—review and editing

## Author ORCIDs

Thomas D James (iD) https://orcid.org/0000-0003-0949-4783
Danielle J Zwiefelhofer (iD) https://orcid.org/0000-0001-7555-2908
C Andrew Frank (iD) https://orcid.org/0000-0001-9599-421X

## Decision letter and Author response

Decision letter https://doi.org/10.7554/eLife.39643.028
Author response https://doi.org/10.7554/eLife.39643.029

# Additional files

## Supplementary files

• Supplementary file 1. Summary electrophysiological data for *Figure 1* (*Plc21C* RNAi experiments). Genotypes and/or conditions are denoted. For GAL4 drivers, '*Pre +Post* Gal4' denotes a genetic combination of *elaV(C155)-Gal4/Y; Sca-Gal4/+; BG57-Gal4/+*. Average values ± SEM are presented for each electrophysiological parameter, with *n* = number of NMJs recorded. Values include miniature excitatory postsynaptic potential (mEPSP) amplitude, mEPSP frequency (Freq), excitatory postsynaptic potential (EPSP) amplitude, quantal content (QC), and QC corrected for non-linear summation (NLS). *$p < 0.05$, **$p < 0.01$, ***$p < 0.001$ vs. unchallenged control.
DOI: https://doi.org/10.7554/eLife.39643.019

• Supplementary file 2. Summary electrophysiological data for *Figure 2* (screen and follow-up). Genotypes and/or conditions are denoted. The data are split into two tables. The first table summarizes the screen data from *Figure 2A*. The second table summarizes the follow-up data examining the *UAS-IP$_3$-sponge* reagent, including the homoeostatic block identified in the screen. Average values ± SEM are presented for each electrophysiological parameter, with *n* = number of NMJs recorded. Values include miniature excitatory postsynaptic potential (mEPSP) amplitude, mEPSP frequency (Freq), excitatory postsynaptic potential (EPSP) amplitude, quantal content (QC), and QC corrected for non-linear summation (NLS). *$p < 0.05$, **$p < 0.01$, ***$p < 0.001$ vs. unchallenged control.
DOI: https://doi.org/10.7554/eLife.39643.020

• Supplementary file 3. Summary electrophysiological data for *Figure 4* (Xestospongin C application). Genotypes and/or conditions are denoted. Average values ± SEM are presented for each electrophysiological parameter, with *n* = number of NMJs recorded. Values include miniature excitatory postsynaptic potential (mEPSP) amplitude, mEPSP frequency (Freq), excitatory postsynaptic potential (EPSP) amplitude, quantal content (QC), and QC corrected for non-linear summation (NLS). *p<0.05, **p<0.01, ***p<0.001 vs. unchallenged control.
DOI: https://doi.org/10.7554/eLife.39643.021

• Supplementary file 4. Summary electrophysiological data for *Figure 5* (2-APB application). Genotypes and/or conditions are denoted. Average values ± SEM are presented for each electrophysiological parameter, with *n* = number of NMJs recorded. Values include miniature excitatory postsynaptic potential (mEPSP) amplitude, mEPSP frequency (Freq), excitatory postsynaptic potential (EPSP) amplitude, quantal content (QC), and QC corrected for non-linear summation (NLS). *p<0.05, **p<0.01, ***p<0.001 vs. unchallenged control.
DOI: https://doi.org/10.7554/eLife.39643.022

• Supplementary file 5. Summary electrophysiological data for *Figure 6* (Ryanodine and Dantrolene applications). Genotypes and/or conditions are denoted. Average values ± SEM are presented for each electrophysiological parameter, with *n* = number of NMJs recorded. Values include miniature excitatory postsynaptic potential (mEPSP) amplitude, mEPSP frequency (Freq), excitatory postsynaptic potential (EPSP) amplitude, quantal content (QC), and QC corrected for non-linear summation (NLS). *p<0.05, **p<0.01, ***p<0.001 vs. unchallenged control.
DOI: https://doi.org/10.7554/eLife.39643.023

• Supplementary file 6. Summary electrophysiological data for *Figure 7* (manipulation interaction analyses). Genotypes and/or conditions are denoted. Average values ± SEM are presented for each electrophysiological parameter, with *n* = number of NMJs recorded. Values include miniature excitatory postsynaptic potential (mEPSP) amplitude, mEPSP frequency (Freq), excitatory postsynaptic potential (EPSP) amplitude, quantal content (QC), and QC corrected for non-linear summation (NLS). *p<0.05, **p<0.01, ***p<0.001 vs. unchallenged control.
DOI: https://doi.org/10.7554/eLife.39643.024

• Supplementary file 7. Summary electrophysiological data for *Figure 8* (tissue specificity analyses). Genotypes and/or conditions are denoted. Average values ± SEM are presented for each electrophysiological parameter, with *n* = number of NMJs recorded. Values include miniature excitatory postsynaptic potential (mEPSP) amplitude, mEPSP frequency (Freq), excitatory postsynaptic potential (EPSP) amplitude, quantal content (QC), and QC corrected for non-linear summation (NLS). *p<0.05, **p<0.01, ***p<0.001 vs. unchallenged control.
DOI: https://doi.org/10.7554/eLife.39643.025

• Transparent reporting form
DOI: https://doi.org/10.7554/eLife.39643.026

### Data availability

All data generated or analysed during this study are included in the manuscript and supporting files. Summary data for electrophysiology are included in the Supplementary Tables. Raw data for all figures are included in the Raw Data Workbook Excel file.

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
