## [Decision Letter]

[**Editorial note:** This article has been through an editorial process in which the authors decide how to respond to the issues raised during peer review. The Reviewing Editor's assessment is that all the issues have been addressed.]

Thank you for submitting your article "Maintenance of homeostatic plasticity at the *Drosophila* neuromuscular synapse requires continuous IP_3_-directed signaling" for consideration by *eLife*. Your article has been reviewed by two peer reviewers, and the evaluation has been overseen by K VijayRaghavan as the Reviewing Editor and Senior Editor. The reviewers have opted to remain anonymous.

The Reviewing Editor has highlighted the concerns that require revision and/or responses, and we have included the separate reviews below for your consideration. If you have any questions, please do not hesitate to contact us.

Summary (From Reviewer #1):

Synaptic homeostasis comprises a number of processes that help to maintain the strength of synaptic transmission at an optimal level through a variety of conditions. One accessible model for this is the *Drosophila* NMJ, at which a reduction in postsynaptic receptor capacity (typically measured as mEPSPs) leads to a compensatory increase in the number of synaptic vesicles released (QC, quantal content), so as to maintain overall synaptic strength. Both acute (pharmacologically induced) and chronic (genetically induced) homeostatic responses to lowered postsynaptic receptor capacity can be observed at the NMJ, but the extent to which they share similar mechanisms has not been extensively investigated.

James et al. present strong evidence that the two responses can be separated – most strikingly that in some circumstances that impair the response to chronic receptor blockage, the response to acute blockage remains intact. They also present evidence for a partial mechanism, using both genetics and pharmacology to implicate release from both presynaptic and postsynaptic ER calcium stores in the mechanism, and also provide evidence that both pre- and post-synaptic mechanisms are involved.

The main strength of the work is that it is potentially a valuable contribution to dissecting the two responses (acute and chronic), and their molecular mechanisms; it should, therefore, be of wide interest, and also suggests directions for future mechanistic work on the area. My main criticism is that the authors' conclusions depend critically on their quantitative and statistical analyses, and these are not currently explicit or complete enough for me to give my wholehearted support to the paper. These shortcomings would, therefore, need to be addressed – I've elaborated in more depth in the section below on data and statistics.

Separate reviews (please respond to each point):

*Reviewer #1:*

Introductory Paragraph of Reviewer 1 is the summary above.

Minor Comments:

Subsection “Pharmacology targeting Ryanodine Receptors uncouples the induction and maintenance of PHP” and Discussion. The authors argue that IP_3_R and RyR function in a common pathway. But if blockage of either abolishes chronic homeostasis, is it not equally plausible that they function independently, but that both are necessary for homeostasis?

Subsection “PLCb- and IP_3_-directed Signaling is Required for PHP Maintenance”: The downstream consequences of blocking RyR and IP_3_R signaling are discussed at length here, but hardly anything about the possible upstream events. And is it plausible that one of these could be acting (more) presynaptically and the other (more) postsynaptically?

Figure 6A. One of the important comparisons should be *GluRIIA* driver control with *GluRIIA* IP_3_-sponge? This comparison is shown for only 3 of the 4 graphs in A and B, but not for the left graph in A. Presumably it should also be shown in A? And why is there a comparison here of driver control with IP_3_ sponge, but not in the other graphs?

Introduction, sentence four: has > have

Introduction paragraph four: previously identified Plc21 – as what?

Introduction paragraph five: *Drosophila* in italics (maybe elsewhere too, I just noticed it here)

Subsection “Pharmacology targeting IP_3_ receptors uncouples the induction and maintenance of PHP”: throughout the soma (including synapses) – soma does not include synapses

Additional data files and statistical comments (important):

1) Instead of showing graphs with mean + SEM superimposed on a bar, graphs should show mean + SEM, or box and whiskers (median and interquartile range, with 5/95 or 10/90 percentiles) for non-parametric distributions, superimposed on all datapoints. Number of datapoints should be stated for each mean + SEM or box/whiskers.

2) Comparisons would be easier to visualize if the same quantities were always graphed together, e.g. all mEPSPs together across genotypes or treatments, or all QCs together, as in Figure 6 – instead of grouping genotypes or treatments, as in other figures.

3) A corollary of points 1 and 2 is that the distribution of control datapoints should also be shown for all graphs, rather than just showing the non-control readings expressed as numbers normalized to controls.

4) A corollary or points 1, 2 and 3 is that all readings in graphs should be expressed as raw units, e.g. mV, not normalized to 100. This is necessary for the reader to assess how the authors' values compare with those of other workers, and even across the paper.

5) The authors also leave a lot up to the faith of the reader by presenting "typical" plots of EPSPs and mini-EPSPs. For a paper where so much hangs by the quantitative comparisons, I'd rather see plots of mean EPSPs (although this may not be possible for mEPSPs).

6) When the authors show graphs of mEPSPs and QC, they should also show an equivalent graph for EPSPs across genotypes or treatments. The authors' conclusions depend critically on their estimates of QC, and because of the non-linear relationship of QC with increasing EPSP magnitude, the correction required makes QC estimate less accurate for large EPSPs than for small ones. This intrinsic inaccuracy also a reason why the authors must do better at showing their raw data and analyses. The ideal solution would be to use 2-electrode voltage clamping to measure mEPSCs and EPSCs, so that no correction is required to estimate QC – although the authors' approach might still be enough if they improve the rigor used to show and explain their measurements and analyses.

7) Since the authors' quantitative analyses are so critical to their conclusions, they should also deposit all their supporting raw datapoints and analyses (not just summarized parameters as in current Supplementary files), either in supplementary info, or with a permanent DOI in a public data repository, linked to the paper. Datapoints should be deposited in a file format that allows any reader to analyze them, e.g. csv/tsv, Excel, Graphpad or a common stats package. GraphPad Prism files are convenient since they also show analyses, but since not all readers may have the relevant software, datapoints should also be provided in a widely accessible format like tsv/csv/Excel.

*Reviewer #2:*

The study by James el al follows up on their previous identification of PLCβ as a key enzyme for maintenance of PHP. Here they dissect the signaling downstream PLCβ and show that IP_3_ and Ryanodine receptor (RyR) are part of the same signaling pathway required to maintain PHP. Interestingly, synapses with impairements in PLCβ, IP_3_ or RyR signaling respond to acute pharmacological challenges (such as PhTox) albeit thay cannot sustain expression of PHP. The authors combine genetics and pharmacological manipulations to document the separation of induction vs. maintenance processes in presynaptic homeostatic plasticity.

Strengths:

1) Molecular pinning of the short-term induction and long-term maintenance processes during PHP. The authors have already contributed to the conceptual separation of the two processes as well as the basis for acute and chronic synaptic homeostasis. Here they take these studies further and describe a signaling pathway and pharmacological tools that differentiate between PHP induction and maintenance.

2) Clever use of genetics and pharmacological manipulations to challenge the system and isolate different responses.

3) The authors show that release of calcium from intracellular stores is required for the maintenance of PHP; this is separate from the presynaptic neurotransmitter release, which is highly sensitive to Ca^2+^ influx through voltage-gated Cav2 channels.

4) They report that PHP adjusts neurotransmission release independent of the mechanisms that set baseline levels.

5) Well written manuscript, with strong arguments and clearly described contributions to the field.

Weaknesses:

1) The authors find that IP_3_ signaling in both neuron and muscle contribute to the long-term homeostatic potentiation, but the underlyining pre- or postsynaptic mechanisms remain unknown. At the least, the athors should measure the RRP size. Ideally, Ca^2+^ influx should also be determined.

2) As described in the text, once PLCβ is activated by Gαq, it cleaves the PIP2 into DAG and IP_3_. In here, the authors only address the role of IP_3_-directed signaling. What about DAG? In *C. elegans*, DAG is thought to recruit UNC-13 at release sites, potentiating synaptic transmission (McMullan et al., Genes Dev. 2006).

3) Previous studies (including by these authors) have demonstrated that motoneuron activity is not required for the rapid induction of synaptic homeostasis. Severing the axon does not disrupt the PHP induction; however, the ER and the RyR activity should be greatly altered. Is the induction of PHP in severed axons less efficient in the absence of PLCβ, IP_3_ or RyR? How do the authors reconcile their observations?

4) Xestospongin-C appears to inhibit voltage-dependent Ca^2+^ and K^+^ currents at a concentration range similar to that at which it inhibits the IP_3_ receptor in intact smooth muscle in rodents (Ozaki et al., 2002). Thus, Xestospongin-C is a selective blocker of the IP_3_ receptor in permeabilized cells but is likely less efficient in cells with intact plasma membrane such as *Drosophila* NMJ. Are there any inhibitory effects of Xestospongin-C on pre- and/or postsynaptic voltage-dependent Ca^2+^ and K^+^ currents at larval NMJ?

5) 2-APB impairs IP_3_R but also other targets such as TRP channels. A TRPV channel, Inactive (Iav), maintains presynaptic resting Ca^2+^ concentration by promoting Ca^2+^ release from the ER in *Drosophila* motor neurons, and is required for both synapse development and neurotransmission (Wong et al., 2014). Did the authors measure any effects of 2-APB on Inactive (or other *Drosophila* TRP channels)? How could they rule out that 2-APB influences the TRP channels-mediated basal neurotransmission and/or synaptic homeostasis?

Minor Comments:

Figure 3. The number of boutons should be reported per muscle area.

---

## [Author Response]

Separate reviews (please respond to each point):

Reviewer #1:

Minor Comments:Subsection “Pharmacology targeting Ryanodine Receptors uncouples the induction and maintenance of PHP” and Discussion. The authors argue that IP_3_R and RyR function in a common pathway. But if blockage of either abolishes chronic homeostasis, is it not equally plausible that they function independently, but that both are necessary for homeostasis?

The reviewer is correct; this is one potential interpretation of the data. There is not necessarily a direct linear link between IP_3_, IP_3_R, and RyR. We revised our text to include this possibility. We also removed a pathway cartoon model that was included in the original submission. In terms of defining a “pathway,” we explain our thinking in more detail here.

By classical analysis, phenotypic non-additivity of two or more similar manipulations is consistent with a common pathway – as long as at least one manipulation is a functional null. Phenotypic additivity that is more severe than the single manipulations is consistent with parallel pathways.

For our study, the output is presynaptic homeostatic potentiation (PHP) – i.e., when the NMJ is challenged, how much the quantal content is potentiated vs. baseline. The raw measures are spontaneous and evoked depolarization. We found that the UAS- IP_3_-sponge expression and 10 µM dantrolene manipulations alone each abrogate the long-term maintenance of PHP. Each manipulation alone is “null” for PHP maintenance. With both manipulations together, there is no further decrease in the level of evoked synaptic neurotransmission versus either manipulation alone. These data suggest that the two manipulations are acting upon the same process and not blunting neurotransmission through divergent processes that are independent of PHP maintenance. This is how we made our “pathway” interpretation.

The reviewer’s point is well taken. Whether that specific form of PHP regulation is through a direct linear pathway or through a convergent pathway is not determined. We revised the text in recognition of this point.

Subsection “PLCb- and IP_3_-directed Signaling is Required for PHP Maintenance”: The downstream consequences of blocking RyR and IP_3_R signaling are discussed at length here, but hardly anything about the possible upstream events. And is it plausible that one of these could be acting (more) presynaptically and the other (more) postsynaptically?

This is a good point. Possible upstream signals include those that activate PLCb, potentially through Gq-coupled GPCRs. In our prior publication (Brusich et al., 2015), we showed that partial loss of Gq function partially impaired homeostatic plasticity. In our revised Discussion section, we include information about potential upstream signals.

The reviewer is correct regarding tissue type. The signaling roles for RyR and IP_3_R could be in separate tissues (or in both tissues). Our data are consistent with dual pre- and postsynaptic functions for IP_3_. Our texts includes all of the possibilities that go along with that data.

Figure 6A. One of the important comparisons should be GluRIIA driver control with GluRIIA IP_3_-sponge? This comparison is shown for only 3 of the 4 graphs in A and B, but not for the left graph in A. Presumably it should also be shown in A? And why is there a comparison here of driver control with IP_3_ sponge, but not in the other graphs?

We thank the reviewer for pointing this out. Those data in the original submission Figure 6 are now in Revision Figure 8. We display parallel comparisons for all panels in Figures 8A-F.

Introduction, sentence four: has > have

Correct, thank you.

Introduction paragraph four: previously identified Plc21 – as what?

Thank you. We previously identified Plc21C as a factor needed to implement PHP. We edited the text.

Introduction paragraph five: Drosophila in italics (maybe elsewhere too, I just noticed it here)

We have italicized all instances of *Drosophila* and *Drosophila melanogaster* in the revision. The exceptions are instances of *Drosophila* that are unitalicized in citation titles and also the unitalicized names of the two organizations from which we acquired many of the fly lines we used (Bloomington *Drosophila* Stock Center and Vienna *Drosophila* Resource Center). For the original submission we had only italicized the instances that included both genus and species names.

Subsection “Pharmacology targeting IP_3_ receptors uncouples the induction and maintenance of PHP”: throughout the soma (including synapses) – soma does not include synapses

Correct, thank you. The sentence should read that the ER is known to localize throughout neurons (including synaptic terminals).

Additional data files and statistical comments (important):1) Instead of showing graphs with mean + SEM superimposed on a bar, graphs should show mean + SEM, or box and whiskers (median and interquartile range, with 5/95 or 10/90 percentiles) for non-parametric distributions, superimposed on all datapoints. Number of datapoints should be stated for each mean + SEM or box/whiskers.

We revised every data figure in response to this comment. We agree that there are more informative ways to visualize data than bar graphs +/- SEM.

We tried plotting box-and-whisker plots superimposed on all data points. When we plotted individual points, it added a visual difficulty – namely, the points (no matter how small we rendered them) frequently obscured important information (e.g. where the 25^th^, 50^th^, and 75^th^ percentiles were). As an alternative, we used violin plots for the revision. The violin plots are composed of lines at the 0^th^, 25^th^, 50^th^, 75^th^, and 100^th^ percentiles. They contain all information that would have been in a box-and-whisker plots. Moreover, the violin shape demonstrates the data distribution, which serves a similar purpose as plotting all points. *n* values are beneath the violins.

In the interest of transparency (and as suggested by the reviewer in a subsequent comment below), we also compiled an Excel sheet reporting raw data for all individual data points for electrophysiological recordings or synapse growth measurements. By including the Excel data sheet with the manuscript, we hope that this serves a similar purpose as would be served by plotting all of those data points. The only figure for which we do not do this is revision Figure 2A. Figure 2A summarizes the screen data. Still, for the screen we report the summary data in table form.

2) Comparisons would be easier to visualize if the same quantities were always graphed together, e.g. all mEPSPs together across genotypes or treatments, or all QCs together, as in Figure 6 – instead of grouping genotypes or treatments, as in other figures.

This is a helpful point. For the revision, we graphed the same measures together across genotypes and treatments.

3) A corollary of points 1 and 2 is that the distribution of control datapoints should also be shown for all graphs, rather than just showing the non-control readings expressed as numbers normalized to controls.

We agree that this is a better way to compare data sets. For the revision, we show the control data sets as violin plots. We also added all individual control data points to the Excel document.

4) A corollary or points 1, 2 and 3 is that all readings in graphs should be expressed as raw units, e.g. mV, not normalized to 100. This is necessary for the reader to assess how the authors' values compare with those of other workers, and even across the paper.

For the revision, we plotted all of the readings as raw units, rather than as relative percentages. The exception is Figure 2A, where a summary of the relative increase in quantal content was needed to assess whether or not a screen candidate expressed PHP.

5) The authors also leave a lot up to the faith of the reader by presenting "typical" plots of EPSPs and mini-EPSPs. For a paper where so much hangs by the quantitative comparisons, I'd rather see plots of mean EPSPs (although this may not be possible for mEPSPs).

We hope that by showing the raw data measures in the revision that we are not leaving anything quantitative up to subjective judgement, either by averages or by statistics.

We thank the reviewer for pointing out a potential point of confusion with regard to electrophysiological traces. Our strategy for picking a representative EPSP trace for a dataset condition has been to analyze NMJs individually and then to calculate averages across all NMJs in the dataset. To pull a trace from a specific recording, we pick a recording that is at (or nearly at) the calculated average. In the case of EPSPs, the final trace that is rendered is an average of all the EPSP recordings from that particular NMJ. We have clarified these methods for the revision.

If we are interpreting the reviewer’s comment correctly, it seems that the preference would be to have our software render a single EPSP trace that represents the entire dataset, instead of just one NMJ. This is possible. For a typical dataset this might be *n* = 15 muscles x 30 EPSPs analyzed per muscle = 450 EPSPs analyzed per genotype. Yet that method would not generate a visual trace that is appreciably different in depolarization than the one that we are already rendering for the “typical” NMJ from the dataset. Moreover, it would not represent any real NMJ recording, but it would be rendering of many recordings.

The reviewer is correct about rendering mEPSPs; an “average of averages” sort of rendering across all muscles is not practical for that measure.

6) When the authors show graphs of mEPSPs and QC, they should also show an equivalent graph for EPSPs across genotypes or treatments. The authors' conclusions depend critically on their estimates of QC, and because of the non-linear relationship of QC with increasing EPSP magnitude, the correction required makes QC estimate less accurate for large EPSPs than for small ones. This intrinsic inaccuracy also a reason why the authors must do better at showing their raw data and analyses. The ideal solution would be to use 2-electrode voltage clamping to measure mEPSCs and EPSCs, so that no correction is required to estimate QC – although the authors' approach might still be enough if they improve the rigor used to show and explain their measurements and analyses.

There are multiple considerations in choosing two-electrode voltage clamp (TEVC) vs. sharp electrode bridge mode recordings of synaptic voltages for this prep. The reviewer is correct – TEVC yields real calculations of QC after measuring currents, while recording in bridge mode yields estimates of QC after measuring synaptic voltages. TEVC also offers information about membrane current kinetics. On the other hand, noise levels of a TEVC recording are far greater than in bridge mode and can lead to consequential inaccuracies in analyzing the mEPSC quanta and the calculated QC. This would be especially true for our recordings in which were concurrently knocking down *GluRIII* gene function and applying PhTox-433.

Our compromise has been to record mEPSPs and EPSPs in an external [Ca^2+^] that is low. This serves to diminish possible effects of non-linear summation (NLS), and it is a common practice in the field. We do take NLS into account by calculating QC, both without a correction and with a correction. This is also common practice if synaptic voltages are recorded. We source the NLS correction method (Martin, 1955) in the revised Materials and methods. The full data (including the uncorrected QCs and corrected QCs) are in the new Excel spreadsheet we have included with the revision.

7) Since the authors' quantitative analyses are so critical to their conclusions, they should also deposit all their supporting raw datapoints and analyses (not just summarized parameters as in current Supplementary files), either in supplementary info, or with a permanent DOI in a public data repository, linked to the paper. Datapoints should be deposited in a file format that allows any reader to analyze them, e.g. csv/tsv, Excel, Graphpad or a common stats package. GraphPad Prism files are convenient since they also show analyses, but since not all readers may have the relevant software, datapoints should also be provided in a widely accessible format like tsv/csv/Excel.

This is a good suggestion. We are providing an Excel data sheet for the final version of the paper. This file will be downloadable by readers who would like to study our raw data.

Reviewer #2:

The study by James el al follows up on their previous identification of PLCβ as a key enzyme for maintenance of PHP. Here they dissect the signaling downstream PLCβ and show that IP_3_ and Ryanodine receptor (RyR) are part of the same signaling pathway required to maintain PHP. Interestingly, synapses with impairements in PLCβ, IP_3_ or RyR signaling respond to acute pharmacological challenges (such as PhTox) albeit thay cannot sustain expression of PHP. The authors combine genetics and pharmacological manipulations to document the separation of induction vs. maintenance processes in presynaptic homeostatic plasticity.Strengths:1) Molecular pinning of the short-term induction and long-term maintenance processes during PHP. The authors have already contributed to the conceptual separation of the two processes as well as the basis for acute and chronic synaptic homeostasis. Here they take these studies further and describe a signaling pathway and pharmacological tools that differentiate between PHP induction and maintenance.2) Clever use of genetics and pharmacological manipulations to challenge the system and isolate different responses.3) The authors show that release of calcium from intracellular stores is required for the maintenance of PHP; this is separate from the presynaptic neurotransmitter release, which is highly sensitive to Ca^2+^ influx through voltage-gated Cav2 channels.4) They report that PHP adjusts neurotransmission release independent of the mechanisms that set baseline levels.5) Well written manuscript, with strong arguments and clearly described contributions to the field.

We thank the reviewer for this overall positive assessment of the manuscript.

Weaknesses:1) The authors find that IP_3_ signaling in both neuron and muscle contribute to the long-term homeostatic potentiation, but the underlyining pre- or postsynaptic mechanisms remain unknown. At the least, the athors should measure the RRP size. Ideally, Ca^2+^ influx should also be determined.

The reviewer is correct. There are tissue-specific aspects of the mechanism that we do not yet know. The comment points to two related presynaptic mechanisms (RRP size, Ca^2+^ influx). Prior studies have documented increases in presynaptic Ca^2+^ transients and increases in RRP size after a homeostatic challenge, like PhTox application (induction) or *GluRIIA* gene mutation (maintenance). Those measures correlate with whether or not PHP induction and maintenance are successful.

There are confounding issues for this particular study. Given these confounds, we believe it is appropriate to leave this mechanistic analysis for future studies. First, as pointed out by the reviewer, multiple mechanisms are likely implemented given the dual pre- and postsynaptic requirement for IP_3_ signaling. Second, to measure Ca^2+^ transients at the NMJ, one usually uses Ca^2+^ indicators that flood the terminal. Given the timescale of evoked release, such indicators would not easily distinguish between influx through Ca_V_2-type channels vs. efflux from stores like the endoplasmic reticulum (ER). It is probable that sequestration of IP_3_ and pharmacological impairment of IP_3_Rs or RyRs affect the latter. Finally, it is not trivial to separate increases in terminal Ca^2+^ from increases in RRP size. By definition, greater influx of extracellular Ca^2+^ will recruit more vesicles to be released.

2) As described in the text, once PLCβ is activated by Gαq, it cleaves the PIP2 into DAG and IP_3_. In here, the authors only address the role of IP_3_-directed signaling. What about DAG? In C. elegans, DAG is thought to recruit UNC-13 at release sites, potentiating synaptic transmission (McMullan et al., Genes Dev. 2006).

When we designed our screen (Figure 2 and Supplementary file 2), we did consider the possible effects of diacyl glycerol (DAG). We included a *UAS-RNAi* line against *unc-13* in our screen, and we also included a *UAS-PKCi* (inhibitory) construct, since DAG is known to activate PKC and affect Ca_V_2 channel activity. In neither case did we observe a block of PHP maintenance. Those data are summarized in Supplementary file 2.

Negative data from a screen do not rule out a molecule or a process. Thus, based on our screening data alone, we agree with the reviewer that the DAG branch of the pathway has not been thoroughly examined. For the remainder of the study, we focused on the IP_3_ branch of the pathway because of our screen results.

Interestingly, new studies that touch upon this issue have been published since our original submission (Goel et al., 2019; Böhme et al., 2019; and Gratz et al., 2019). These new studies indicate that active zone proteins like UNC-13A, Cacophony (Cac), and Bruchpilot (Brp) scale with release, by immunostaining or by examining endogenously tagged Cac. We cite these new studies in our revision.

Those published results do not necessarily mean that one would expect *unc-13[RNAi]* or *cac[RNAi]* to yield a positive hit in a homeostasis screen like ours. That is because knockdown of gene function by RNAi is a not a null condition. It should still leave residual wild-type UNC-13A protein around. That residual wild-type UNC-13A could theoretically be scaled with homeostatic need. We previously addressed with issue with *cac*. Knock down by RNAi of Ca_V_2/*cacophony* does not prevent PHP maintenance (Brusich et al. *Frontiers in Cell. Neuro*), even though Cac protein levels scale with PHP (Goel et al., 2019; Böhme et al., 2019 and Gratz et al., 2019).

3) Previous studies (including by these authors) have demonstrated that motoneuron activity is not required for the rapid induction of synaptic homeostasis. Severing the axon does not disrupt the PHP induction; however, the ER and the RyR activity should be greatly altered. Is the induction of PHP in severed axons less efficient in the absence of PLCβ, IP_3_ or RyR? How do the authors reconcile their observations?

The reviewer’s experiment is interesting and not previously tested to our knowledge. To test for a possible synergistic interaction between severing axons, IP_3_-directed signaling, and ER function during the acute induction phase of PHP, we applied PhTox to NMJs with intact motor nerves or with cut motor nerves; moreover, we did this in a genetic background while expressing the *UASIP_3_-sponge* construct pre- and postsynaptically (drivers alone for control). The result is that PHP induction still works in the *UAS-IP_3_-sponge*-expressing background, regardless of whether or not the motor nerve is severed during drug application (Figure 7).

This result is consistent with the prior work. Motor nerve severing does not impair PHP induction by PhTox (Frank et al., 2006); IP_3_ sequestration does not prevent PHP induction by PhTox with an intact preparation (present study, revision Figures 2, 7); and motor nerve severing plus IP_3_ sequestration does not prevent PHP induction with a severed nerve preparation (present study, revision Figure 7). Combined motor nerve severing and IP_3_ sequestration may make the process a little less efficient (Figure 7), but PHP is still present.

4) Xestospongin-C appears to inhibit voltage-dependent Ca^2+^ and K^+^ currents at a concentration range similar to that at which it inhibits the IP_3_ receptor in intact smooth muscle in rodents (Ozaki et al., 2002). Thus, Xestospongin-C is a selective blocker of the IP_3_ receptor in permeabilized cells but is likely less efficient in cells with intact plasma membrane such as Drosophila NMJ. Are there any inhibitory effects of Xestospongin-C on pre- and/or postsynaptic voltage-dependent Ca^2+^ and K^+^ currents at larval NMJ?

We thank the reviewer for pointing this out. We added this additional reference to our text in order to report known effects of Xestospongin C.

For the *Drosophila* NMJ, we do not have direct data addressing this particular question. However, we do know that impairment of voltage-dependent Ca^2+^ and K^+^ currents would likely have profound effects on baseline neurotransmission. Our data argue against the idea: Xestospongin C application does not have appreciable effects on baseline presynaptic release (e.g. revision Figure

4).

In principle, Xestospongin C could be affecting presynaptic or postsynaptic parameters, independent of the IP_3_ receptors. There potential issues with specificity or selectivity with any pharmacological reagent. This is why we tested multiple reagents, in addition to compiling genetic lines of evidence.

5) 2-APB impairs IP_3_R but also other targets such as TRP channels. A TRPV channel, Inactive (Iav), maintains presynaptic resting Ca^2+^ concentration by promoting Ca^2+^ release from the ER in Drosophila motor neurons, and is required for both synapse development and neurotransmission (Wong et al., 2014). Did the authors measure any effects of 2-APB on Inactive (or other Drosophila TRP channels)? How could they rule out that 2-APB influences the TRP channels-mediated basal neurotransmission and/or synaptic homeostasis?

The reviewer is correct. We had similar thoughts – a *priori*, we thought that Inactive could be a potential signaling component, so we included it in the screen (see Supplemental file). The screen did not produce data suggesting that Inactive is required for PHP maintenance. With the caveat that a negative screen result would not necessarily rule out a factor’s involvement, our data do not support the idea that 2-APB is exerting its effects on PHP through Inactive. For completeness, we cite the reference mentioned by the reviewer in our revision.

Minor Comments:Figure 3. The number of boutons should be reported per muscle area.

We include this measurement for the revision Figure 3.